

# A reduced-order Kalman smoother for (paleo-)ocean state estimation: assessment and application to the LGM

Charlotte Breitkreuz[1], André Paul[1], Stefan Mulitza[1], Javier García-Pintado[1], and Michael Schulz[1]

[1]MARUM - Center for Marine Environmental Sciences and Faculty of Geosciences, University of Bremen, Bremen, Germany

**Correspondence:** Charlotte Breitkreuz (cbreitkreuz@marum.de)

**Abstract.** Combining ocean models and proxy data via data assimilation is a powerful means to obtain more reliable estimates of past ocean states, but studies using data assimilation for paleo-ocean state estimation are rare. A few studies used the adjoint method, also called 4D-Var, to estimate the state of the ocean during the Last Glacial Maximum (LGM). The adjoint method, however, requires the adjoint of the model code, which is not easily obtained for most models. The method is computationally very demanding and does not readily provide uncertainty estimates. Here, we present a new and computationally very efficient technique to obtain ocean state estimates. We applied a state reduction approach in conjunction with a finite difference sensitivity-iterative Kalman smoother (FDS-IKS) to estimate spatially varying atmospheric forcing fields and to obtain an equilibrium model simulation in consistency with proxy data. We tested the method in synthetic pseudo-proxy data experiments. The method is capable of very efficiently estimating 16 control variables and reconstructing a target ocean circulation from sea surface temperature (SST) and oxygen isotopic composition of seawater data at LGM coverage. The method is advantageous over the adjoint method regarding that it is very easy to implement, it requires substantially less computing time and provides an uncertainty estimate of the estimated control variables. The computing time, however, depends linearly on the size of the control space limiting the number of control variables that can be estimated. We used the method to investigate the constraint of data outside of the Atlantic Ocean on the Atlantic overturning circulation. Our results indicate that while data from the Pacific or Indian Ocean aid in correctly estimating the Atlantic overturning circulation, they are not as crucial as the Atlantic data. We additionally applied the method to estimate the LGM ocean state constrained by a global SST reconstruction and data on the oxygen isotopic composition of calcite from fossil benthic and planktic foraminifera. The LGM estimate shows a large improvement compared to our first guess, but model-data misfits remain after the optimization due to model errors that cannot be corrected by the control variables. The estimate shows a shallower North Atlantic Deep Water and a weaker Atlantic overturning circulation compared to today in consistency with previous studies. The combination of the FDS-IKS and the state reduction approach is a step forward in making ocean state estimation and data assimilation applicable for complex and computationally expensive models and to models where the adjoint is not available.

*Copyright statement.* TEXT



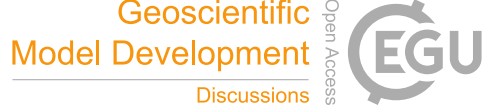

## 1 Introduction

Information on past climate states are usually obtained from proxy data or through climate model simulations. Data assimilation methods combine the information from both sources and can help to obtain more reliable estimates of past climate states. A particularly interesting time period is the Last Glacial Maximum (LGM, $19-23$ ka). It was the last time during Earth's history

when the climate was substantially different from today but remained relatively stable for a few thousand years (Clark et al., 2009). Model studies typically use equilibrium simulations to approximate the LGM climate to minimize the influence of unknown initial conditions and because transient boundary conditions are not well known (e.g., Otto-Bliesner et al., 2007). In equilibrium model simulations it is the model parameters and boundary conditions rather than the initial conditions that determine the climate model state. Whereas in the field of numerical weather prediction data assimilation usually aims at

estimating initial conditions, for the LGM climate it is reasonable to consider uncertain model parameters and/or boundary conditions as control variables.

The LGM has been the target in several state estimation studies using the adjoint method to combine general circulation models (GCMs) and proxy data (Dail and Wunsch, 2014; Kurahashi-Nakamura et al., 2017; Amrhein et al., 2018). The adjoint method has the unique advantage that the computing time does not depend strongly on the number of control variables. In the

15 previously mentioned studies fields of atmospheric forcing are estimated that correspond to a number of control variables on the order of $10^6$ to $10^7$. The adjoint method, however, has major drawbacks. First, the "adjoint" of the model code, which is the prerequisite for applying the adjoint method, is not easily obtained for most models. It can either be coded by hand, which is similar in effort to coding the forward model itself, or it can be obtained by "automatic differentiation", which is only possible if the model code is specifically tailored for this application. Second, the adjoint method does not readily provide information

on the uncertainty of the estimated control variables, and finally, the adjoint method is computationally very demanding. Especially, if one aims at long, preferably equilibrium, model simulations, the adjoint method is not very suitable because of the immense computing costs (e.g., Kurahashi-Nakamura et al., 2017; Breitkreuz et al., 2018). To our knowledge, the longest ocean state estimates including the assimilation of data from the deep ocean cover 400 model years (Kurahashi-Nakamura et al., 2017; Breitkreuz et al., 2018), which is not enough to equilibrate tracers in the deep ocean (Wunsch and Heimbach,

2008).

The size of the control space, that is, the number of control variables, can be very high in applications of the adjoint method. But currently it seems not possible to estimate a similarly large number of control variables in a reasonable amount of time without an adjoint. Additionally, in paleo applications the number of observations is usually very small and estimating a large number of control variables leads to an underdetermined problem.

Other parameter estimation methods that are computationally efficient enough to apply with GCMs are rare. Annan et al. (2005a) and Annan et al. (2005b) used a sequential scheme based on the augmented state ensemble Kalman filter to estimate 12 parameters with a coupled model of intermediate complexity and with an atmospheric GCM. The very simple and computationally fast Green's function method was applied with GCMs and has shown very positive results (Menemenlis et al., 2005; Nguyen et al., 2011; Ungermann et al., 2017). Most recently García-Pintado and Paul (2018) proposed the finite difference

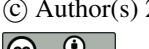


sensitivity-iterative Kalman smoother (FDS-IKS) and the finite difference sensitivity-multistep Kalman smoother (FDS-MKS) for climate state estimation with a comprehensive Earth system model focusing on model parameters as control variables. In their synthetic pseudo-proxy experiments the FDS-IKS has shown better results than the FDS-MKS and the ETKF (García-Pintado and Paul, 2018). The FDS-IKS can be seen as an extension of the Green's function method to more iterations to

account for non-linear sensitivities, or as a reduced alternative to the iterative ensemble Kalman smoother (Bocquet and Sakov, 2014) for low-dimensional control spaces. With the above mentioned ensemble or finite difference sensitivity methods the computational demand rises at least linearly with the number of control variables (García-Pintado and Paul, 2018).

Here, we combine the FDS-IKS with a state reduction approach to estimate the spatially varying atmospheric fields used to force an ocean general circulation model. The FDS-IKS is highly parallelizable and only a few sequential iterations are

needed. It is, therefore, exceptionally fast and it additionally provides an estimate of the uncertainty of the control variables. The state reduction approach aims at reducing the size of the control space. Instead of estimating the atmospheric forcing for each surface grid cell, as it is usually done with the adjoint method, we approximate the meridional mean forcing field by a linear combination of Legendre polynomials and estimate the coefficients of the linear combination. With this approach the size of the control space is reduced dramatically and patchy adjustments of the atmospheric forcing, as obtained with the

adjoint method, are avoided. First, we test the method by applying it to synthetic pseudo-proxy data sampled from target model runs. Second, we use the pseudo-proxy experiments to investigate the constraint placed by data at LGM coverage on the global circulation. In particular, we conduct experiments without data from the Atlantic Ocean and without data from outside of the Atlantic Ocean to infer how large the impact of the data from the different areas of the global ocean on the Atlantic overturning circulation is. Third, we apply it to estimate the LGM ocean state using the Multiproxy Approach for the Reconstruction of

the Glacial Ocean Surface (MARGO) sea surface temperature (SST) reconstruction (MARGO Project Members, 2009) and a compilation of oxygen isotopic composition of calcite ($\delta^{18}O_c$) data from benthic and planktic foraminifera from different sources.

## 2  Material and Methods

### 2.1  Model

We used the Massachussets Institute of Technology general circulation model (MITgcm) in a configuration that solves the Boussinesq form of the hydrostatic Navier-Stokes equations (Marshall et al., 1997; MITgcm Group, 2016). The model includes a non-linear free surface and a rescaled height coordinate (Adcroft and Campin, 2004). A cubed-sphere grid (Ronchi et al., 1996) was used with $192 \times 32$ horizontal grid cells encompassing the global ocean, resulting in a horizontal resolution of about $285 \, \mathrm{km}$, and 15 vertical levels with a resolution of $50 \, \mathrm{m}$ at the surface to $690 \, \mathrm{m}$ at the deepest level reaching a maximum

depth of $5{,}200 \, \mathrm{m}$. The cubed-sphere grid avoids converging grid cells at the poles and, hence, polar singularities. The boundary conditions at the ocean surface were prescribed as monthly atmospheric forcing, namely air temperature, meridional and zonal wind velocities, specific humidity, precipitation, downward shortwave radiation, downward longwave flux and river run-off, based on a fully coupled LGM simulation of the Community Climate System Model Version 3 (Merkel et al., 2010). Bulk





formulae were used to compute outgoing radiation, wind stress and evaporation (Large and Yeager, 2004). The ocean model is fully coupled to a dynamic-thermodynamic sea-ice model (Losch et al., 2010), which used the same grid and the same atmospheric forcing. Subgrid-scale mixing was parameterized with a GM/Redi scheme (Redi, 1982; Gent and McWilliams, 1990). Following Völpel et al. (2018), we excluded the Mediterranean Sea from our model domain because a shallow passage

through the Strait of Gibraltar was not possible due to the coarse vertical resolution. The MITgcm is extended with a water isotope module that comprises the fractionation processes of water isotopes during evaporation over the ocean and allows the model to simulate stable water isotopes in the entire water column (Völpel et al., 2017). In this study, the concentrations of the stable isotopes $H_2^{16}O$ and $H_2^{18}O$ were included in the simulation such that the ratio of the heavier to the lighter isotope in seawater could be computed according to the definition

$$\delta^{18}O_{sw} = \left( \frac{{}^{18}O/{}^{16}O}{R_{VSMOW}} - 1 \right) \cdot 1,000\%o$$

with respect to the Vienna Standard Mean Ocean Water (VSMOW, $R_{VSMOV} = 2,005.2 \cdot 10^{-6}$, Gonfiantini, 1978). The monthly isotopic composition of precipitation and water vapor needed to be prescribed as boundary conditions for the water isotope package and were obtained from a water isotope-enabled simulation with the Community Atmosphere Model version 3.0 (IsoCAM3.0, Tharammal et al., 2013). To avoid drift in the mean salinity and the mean content of the isotopic tracers, we used

a balancing scheme that multiplies the precipitation and isotopic content of precipitation fields by a yearly correction factor (Völpel et al., 2017).

### 2.2 State Reduction Approach

Atmospheric fields used to force ocean models usually originate from model simulations or reanalysis products. They are subject to model errors and other uncertainties, especially for simulations of past climate states. Adjusting atmospheric forcing

fields to fit ocean models towards observational data is common in (paleo-)oceanography (e.g., Köhl and Stammer, 2008; Forget et al., 2015; Kurahashi-Nakamura et al., 2017). At our model grid resolution, one forcing field, for example, air temperature or precipitation, contains $192 \times 32$ variables for each month resulting in 73728 variables per field. We used a state reduction approach to reduce the degrees of freedom and to make it possible to estimate the spatially varying atmospheric forcing with a simple data assimilation technique without the need for an adjoint. Instead of estimating the value in each surface grid cell

for each atmospheric forcing field, we approximated each meridional mean forcing field by a linear combination of Legendre polynomials and estimated the coefficients of the linear combinations as described below. The size of the control space is reduced considerably with this approach. Instead of estimating a number of variables at the order of $10^6$ we estimated a number of variables at the order of $10^1$.

Consider an uncertain first guess forcing field $F$ (e.g., air temperature or precipitation). As first step, we computed the mean

for each $1°$ zonal band at latitude $\varphi$. This meridional mean was approximated by a linear combination of the first three Legendre polynomials

$$\mathcal{P}_F(\varphi) = a_0 P_0(\varphi) + a_1 P_1(\varphi) + a_2 P_2(\varphi),$$





where the Legendre polynomials are

$$P_0(\varphi) = 1$$
$$P_1(\varphi) = \varphi$$
$$P_2(\varphi) = \frac{1}{2}(3\varphi^2 - 1)$$

and $a_0, a_1, a_2$ are scalar coefficients. The coefficients were obtained by fitting the sum of the three first Legendre polynomials to the meridional mean. The original forcing field $F(\varphi, \psi)$ at each grid cell with latitude $\varphi$ and longitude $\psi$ can then be expressed by the sum of the linear combination and a residual term

$$F(\varphi, \psi) = \mathcal{P}_F(\varphi) + r_F(\varphi, \psi). \tag{1}$$

The first three Legendre polynomials $P_0, P_1$, and $P_2$ correspond to the global mean, the north-to-south gradient and the high-to-low latitudinal gradient of the polynomial $\mathcal{P}_F$. Adjusting the coefficients $a_0$, $a_1$ and $a_2$ corresponds to adjusting these quantities in the forcing field. An adjusted forcing field $F_i$ based on coefficients $a_{0,i}$, $a_{1,i}$ and $a_{2,i}$ is computed according to Eq. (1) by adding the original residual term $r_F$ to the perturbed polynomial

$$F_i(\varphi, \psi) = \mathcal{P}_{F,i}(\varphi) + r_F(\varphi, \psi)$$
$$= a_{0,i}P_0(\varphi) + a_{1,i}P_1(\varphi) + a_{2,i}P_2(\varphi) + r_F(\varphi, \psi).$$

In the same way the approach can be applied to the atmospheric forcing field above a certain ocean basin, for example, over the Atlantic Ocean. To allow for more horizontal variation in the forcing fields, we estimated coefficients for the respective forcing fields over the Atlantic Ocean and over the combined Indian and Pacific Ocean. We applied a smoothing around the borders of the two parts of atmospheric forcing to avoid unrealistic steps in the forcing field.

### 2.3 Data Assimilation Method

In this study, we used a simple data assimilation technique to estimate the coefficients of the Legendre polynomials. It is based on the equations of the iterative Kalman smoother (IKS) (e.g., Bell, 1994) and an approximation of the Jacobian of the model operator through finite difference sensitivity experiments. The method was previously used and termed finite difference sensitivities-iterative Kalman smoother (FDS-IKS) by García-Pintado and Paul (2018). First, the iterative Kalman smoother is revisited and subsequently, the approximation of the Jacobian is described.

### Iterative Kalman Smoother

Consider a model operator $\mathcal{G}$ that maps the control variables $\boldsymbol{\lambda} \in \mathbb{R}^p$ from the control space onto the observation space such that the observations $\boldsymbol{y} \in \mathbb{R}^m$ can be described as

$$\boldsymbol{y} = \mathcal{G}(\boldsymbol{\lambda}) + \boldsymbol{\epsilon}, \tag{2}$$





where $\epsilon$ is the observation error that includes measurement and representativeness errors. The equation is based on the "strong-constraint" assumption that the model does not include any errors besides those in the uncertain control variables $\boldsymbol{\lambda}$. Note that in the field of data assimilation typically two operators are considered, one model operator $\mathcal{M}$ mapping from the initial state and the control variables onto the model state and an observation operator $\mathcal{H}$ that maps the model state onto the observation space.

Here, for simplicity of notation and because the strong-constraint assumption is used, the two operators can be summarized by the function composition $\mathcal{G}(\cdot) = \mathcal{H}(\mathcal{M}(\cdot))$.

Let $\boldsymbol{\lambda}^b$ be the first guess of the control variables and $\mathbf{P}^b$ their error covariance matrix ($b$ for "background"). The goal is then to find values for the control variables that minimize the cost function

$$J(\boldsymbol{\lambda}) = \underbrace{\frac{1}{2} (\boldsymbol{y} - \mathcal{G}(\boldsymbol{\lambda}))^{\top} \mathbf{R}^{-1} (\boldsymbol{y} - \mathcal{G}(\boldsymbol{\lambda}))}_{J_{\mathrm{misift}}} + \underbrace{\frac{1}{2} (\boldsymbol{\lambda} - \boldsymbol{\lambda}^b)^{\top} (\mathbf{P}^b)^{-1} (\boldsymbol{\lambda} - \boldsymbol{\lambda}^b)}_{J_{\mathrm{ctrl}}}, \tag{3}$$

where $\mathbf{R}$ is the observation error covariance matrix. The first part of the cost function $J_{\mathrm{misift}}$ measures the model-data misfit. The second part $J_{\mathrm{ctrl}}$ constrains the deviation from the first guess of the control variables. Assuming Gaussian errors for the observations and the first guess control variables, and when $\mathcal{G}$ is linear, the values that minimize the cost function are identical to the "maximum likelihood estimator" and the "minimum variance estimator" (e.g., Evensen, 2009). In this case, this exact solution can be found through the Kalman smoother equations. In most cases, however, the model operator is non-linear. Then

$J$ is not necessarily convex and an exact solution cannot be found (e.g., Van Leeuwen and Evensen, 1996). Instead, the IKS minimizes a sequence of quadratic cost functions to approximate the solution. To that end, $J$ is approximated in an environment of the current best guess of the control variables $\boldsymbol{\lambda}_l$ by a quadratic cost function. In the first iteration the best guess is the first guess, that is, $\boldsymbol{\lambda}_l = \boldsymbol{\lambda}^b$. Consider a small environment $\delta\boldsymbol{\lambda} = \boldsymbol{\lambda} - \boldsymbol{\lambda}_l$ around the current best guess $\boldsymbol{\lambda}_l$. The "tangent linear hypothesis" (TLH)

$$\mathcal{G}(\boldsymbol{\lambda}) - \mathcal{G}(\boldsymbol{\lambda}_l) \approx \mathbf{G}_l (\boldsymbol{\lambda} - \boldsymbol{\lambda}_l) \tag{4}$$

is made, where $\mathbf{G}_l = \mathbf{G}(\boldsymbol{\lambda}_l)$ is the Jacobian or "Tangent Linear Model" of $\mathcal{G}$ evaluated at $\boldsymbol{\lambda}_l$. The TLH assumes that the projection of a small perturbation in the control variables onto a perturbation in model space can be approximated through a linear operator, the derivative of the model at this point in control space. A linear quadratic approximation of the cost function (3) around the current best guess $\boldsymbol{\lambda}_l$ can then be obtained using the TLH (4) as in

$$J_l(\delta\boldsymbol{\lambda}) = \frac{1}{2} (\delta\boldsymbol{y_l} - \mathbf{G}_l \, \delta\boldsymbol{\lambda})^{\top} \mathbf{R}^{-1} (\delta\boldsymbol{y_l} - \mathbf{G}_l \, \delta\boldsymbol{\lambda}) + \frac{1}{2} (\delta\boldsymbol{\lambda} - (\boldsymbol{\lambda}^b - \boldsymbol{\lambda}_l))^{\top} (\mathbf{P}^b)^{-1} (\delta\boldsymbol{\lambda} - (\boldsymbol{\lambda}^b - \boldsymbol{\lambda}_l)) \tag{5}$$

with $\delta\boldsymbol{y_l} = \boldsymbol{y} - \mathcal{G}(\boldsymbol{\lambda}_l)$. The minimum of this quadratic cost function provides a perturbation $\delta\boldsymbol{\lambda}$ that can then be used to compute the new best guess $\boldsymbol{\lambda}_{l+1} = \boldsymbol{\lambda}_l + \delta\boldsymbol{\lambda}$. Assuming Gaussian error statistics, the minimum can be obtained from computing the maximum likelihood/minimum variance estimator via

$$\delta\boldsymbol{\lambda}_l^* = \boldsymbol{\lambda}^b - \boldsymbol{\lambda}_l + \mathbf{K}_l \left( \boldsymbol{y} - \mathcal{G}(\boldsymbol{\lambda}_l) - \mathbf{G}_l \left( \boldsymbol{\lambda}^b - \boldsymbol{\lambda}_l \right) \right), \tag{6}$$

where

$$\mathbf{K}_l = \mathbf{P}^b \left( \mathbf{G}_l \right)^{\top} \left( \mathbf{G}_l \, \mathbf{P}^b \left( \mathbf{G}_l \right)^{\top} + \mathbf{R} \right)^{-1}$$

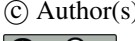


is the Kalman gain (Bell, 1994). Adding the perturbation (6) to the current best guess $\boldsymbol{\lambda}_l$ gives

$$
\begin{aligned}
\boldsymbol{\lambda}_{l+1} &= \boldsymbol{\lambda}_l + \boldsymbol{\delta}\boldsymbol{\lambda}_l^* \\
&= \boldsymbol{\lambda}^b + \mathbf{K}_l \left( \boldsymbol{y} - \mathcal{G}(\boldsymbol{\lambda}_l) - \mathbf{G}_l \left( \boldsymbol{\lambda}^b - \boldsymbol{\lambda}_l \right) \right).
\end{aligned}
\tag{7}
$$

The error covariance matrix of the estimated control variables can be obtained from

$$
\mathbf{P}_{l+1} = (\mathbf{I} - \mathbf{K}_l \mathbf{G}_l) \, \mathbf{P}^b
\tag{8}
$$

(Bell, 1994).

**Finite Difference Sensitivity-Iterative Kalman Smoother**

The matrix $\mathbf{G}_l$ is the Jacobian of the model around the control variables $\boldsymbol{\lambda}_l$. In many applications it is approximated from an ensemble of model runs leading to the ensemble Kalman smoother or ensemble Kalman filter (e.g., Evensen and Van Leeuwen, 2000; Evensen, 2009). Here, the initial conditions were not included in the estimation process and due to the state reduction approach, the control space is comparatively small. This enabled us to use a finite difference sensitivity approach to approximate $\mathbf{G}_l$ (García-Pintado and Paul, 2018). It is less computationally expensive and does not suffer from spurious correlations as ensemble representations of $\mathbf{G}_l$ frequently do when the number of ensemble members is limited (Evensen, 2009).

In each iteration $l$ and for each control variable $\boldsymbol{\lambda}_{l,i}$, $i = 1, ..., p$, we performed three sensitivity runs with a slightly perturbed best guess control variable $\boldsymbol{\lambda}_{l,i}$. The perturbations are random numbers drawn from $\mathcal{N}(0, \sigma_i/100)$ where $\sigma_i$ is the assumed uncertainty of $\boldsymbol{\lambda}_{l,i}$. The other control variables were held fix in these three sensitivity experiments. Therefore, for each iteration $(3 \cdot p) + 1$ runs (three sensitivity runs per control variable and one base line run with the current best guess control variable) were required, which could be performed in parallel. Subsequently, one linear regression for each control variable with the 3+1 experiments and respective model output at each observation location was performed. The slopes of the linear regressions approximate the tangent linear derivative, that is, the Jacobian of the model, $\mathbf{G}_l \in \mathbb{R}^{m \times p}$. For example, the linear regression for control variable $\boldsymbol{\lambda}_{l,i}$, $1 \leq i \leq p$, and observation location $1 \leq j \leq m$ provides the matrix entry $\mathbf{G}_{l_{j,i}}$. Using this approximation of $\mathbf{G}_l$ in the Eq. (7) and (8) leads to the FDS-IKS.

### 2.4 Experimental Design

To test the method, we used pseudo-proxy data to reconstruct target model runs, which represent a surrogate "true" ocean state. The pseudo-proxy data were generated from the target model runs at a realistic LGM data coverage and with realistic LGM uncertainties. We present two sets of pseudo-proxy data experiments. In the first set, the strong-constraint assumption holds true, and thus, the only difference between the target model run and the first guess originate from different values of the control variables that were estimated in the optimization process. These experiments aim at validating the method without being compromised by model errors. In the second set, the strong-constraint assumption does not hold true and the target state was altered by a freshwater hosing that is not a control variable in the optimization. This set of experiments aim at testing the



method in a more realistic setting. In real applications the strong-constraint almost never holds true as model simulations are always influenced by additional model errors.

Each set consists of three experiments where the first experiment used the same data coverage as is available for the LGM, the second used pseudo-proxy data only from the Atlantic Ocean, and the third used data only from the Pacific and Indian Ocean. The two different sets aim at investigating the skill of the method under different conditions, and the three experiments within one set aim at investigating the importance of data outside of the Atlantic Ocean for constraining the Atlantic overturning circulation. Subsequently, the method was applied with reconstructed LGM SST and $\delta^{18}O_c$ data. To create a first guess, the starting point for all experiments, we spun the model up for 3,000 years and continued the spin-up for another 3,000 years including the isotopic tracers reaching a quasi-steady state. All experiments were conducted two times to assess the influence of the random numbers used for generating the perturbations of the control variables. For all experiments the same results, except for insignificant differences, were obtained for both trials and the results described in the following sections are representative for both trials. In each experiment, we performed a total of ten iterations. Each iteration consists of 2,000-year long simulations and the mean of the last 100 years of the respective iteration were used for analysis.

### 2.4.1 Pseudo-proxy Experiments

**Set 1**

In the first set of experiments the strong-constraint assumption holds true and with the correct values for the control variables the model is able to fit the data perfectly within the uncertainties of the observations. We chose 16 control variables that we consider to have a large influence on the simulated ocean circulation and on the simulated oxygen isotopic distribution (Table 1). To generate a target run, we chose values for the corresponding 16 Legendre polynomial coefficients in a random fashion (Target 1, Table 2 and Fig. 1) and performed a 3,000-year run with these target values for the control variables starting from the first guess spin-up. From this target run we sampled $\delta^{18}O_{sw}$ and SST data at the respective locations where $\delta^{18}O_c$ and SST data are available in our LGM data set (Fig. 2). To emulate the uncertainties of the reconstructed LGM data, we randomly added or subtracted a random error following a lognormal distribution with mean and standard deviation of 2.33 °C and 0.89 °C for the SST data, and 0.22 ‰ and 0.13 ‰ for the $\delta^{18}O_{sw}$ data. These numbers were obtained from fitting a lognormal distribution to the uncertainty distribution of the reconstructed LGM data. The assumption of a lognormal distribution disagrees with the assumption of normally distributed errors with mean of zero. Even though the assumption is necessary for the theoretical derivation of the Kalman equations, it is hardly ever fulfilled in applications. In particular, the uncertainties of the LGM proxy data do not follow a Gaussian distribution with mean zero but rather a lognormal distribution. Three experiments were performed that aim at reconstructing Target 1. The first used all generated pseudo-proxy data (P1-All), the second used only the data from within the Atlantic Ocean (P1-Atlantic), and the third used only the data from outside of the Atlantic Ocean (P1-Pacific).



**Set 2**

In the second set of experiments the strong-constraint assumption does not hold true. To generate the target run, we applied a continuous freshwater hosing of $0.24\,\mathrm{Sv}$ in the North Atlantic Ocean between $20°\,\mathrm{N}$ and $50°\,\mathrm{N}$, and additionally changed the 8 control variables related to the isotopic composition in precipitation and water vapor (Target 2, Table 2 and Fig. 1). In the optimization, the previous 16 control variables were estimated, including the 8 control variables for the isotopic composition of precipitation and water vapor. The freshwater hosing was not applied in the first guess or the optimization and the missing hosing emulates a model error that is not accounted for in the optimization. Pseudo-proxy data were generated in the same way from the target run as in Set 1. As above three experiments were performed that aim at reconstructing Target 2, one with all data (P2-All), one with only data from the Atlantic Ocean (P2-Atlantic), one with only data outside of the Atlantic Ocean (P2-Pacific).

### 2.4.2 LGM Experiment

We applied the method to estimate the state of the ocean during the LGM constrained by global SST and $\delta^{18}\mathrm{O_c}$ data from benthic and planktic foraminifera. In the LGM experiment, we used LGM−Late Holocene (LH) anomalies for the model-data comparison. The use of anomalies enables us to disregard species-specific constant offsets from equilibrium $\delta^{18}\mathrm{O_c}$ and we hope that systematic model errors canceled out when anomalies were computed. To compute the simulated anomalies, we used the state estimate of the modern ocean of Breitkreuz et al. (2018). They used the adjoint method to fit the MITgcm in a very similar configuration as is used here, to modern climatological salinity, temperature, and $\delta^{18}\mathrm{O_{sw}}$ data. To compute the isotopic composition of calcite ($\delta^{18}\mathrm{O_c}$) from simulated $\delta^{18}\mathrm{O_{sw}}$ and simulated temperature ($T$ in °C), we used the paleo-temperature equation by Shackleton (1974)

$$T = 16.9 - 4.38 \left(\delta^{18}\mathrm{O_c} - \delta^{18}\mathrm{O_{sw}}\right) + 0.10 \left(\delta^{18}\mathrm{O_c} - \delta^{18}\mathrm{O_{sw}}\right)^2.$$

Beforehand, the simulated $\delta^{18}\mathrm{O_{sw}}$ values were converted from VSMOW to Vienna Peedee belemnite standard (VPDB) by subtracting $0.27\,‰$ (Hut, 1987) and to account for the ice sheet effect a constant value of $1.1\,‰$ was added (Duplessy et al., 2002).

### 2.5 LGM Proxy Data and Uncertainties

To constrain the LGM estimate, we used reconstructed SST and $\delta^{18}\mathrm{O_c}$ data from benthic and planktic foraminifera. The most comprehensive LGM SST anomalies reconstruction data set to date is that of the MARGO Project Members (2009). They used multiple microfossil-based (assemblages of planktic foraminifera and dinoflagellate cysts) and geochemical (Mg/Ca and alkenones) proxies to reconstruct annual SST's. The MARGO Project Members (2009) additionally provide estimates of the respective uncertainties of the reconstructed temperatures and a mean reliability index. The MARGO data, however, show some inter-proxy discrepancies that are not explained by the given uncertainties (MARGO Project Members, 2009; Dail and Wunsch, 2014; Caley et al., 2014). In particular, dinoflagellate cysts and alkenones north of $40°\,\mathrm{N}$ yield higher





temperatures than the foraminiferal assemblages in this region. Dail and Wunsch (2014) list a number of uncertainties related to temperatures reconstructed from dinoflagellate cysts and alkenones at high latitudes that limit our trust in these SST's. To take these uncertainties into account during the optimization, we used the $2\sigma$ uncertainties for the SST values reconstructed from dinoflagellate cysts and alkenones north of $40°$ N as proposed by Dail and Wunsch (2014).

5    For the optimization we averaged the raw data (not the block averages) onto the surface level of our model grid $(0-50\,\mathrm{m})$ resulting in 369 filled grid cells. When multiple values landed in one grid box, we computed the weighted mean according to the reliability index as proposed by the MARGO Project Members (2009). The respective uncertainties were computed accordingly as the sum of the weighted grid cell mean uncertainty and the standard deviation of the values within one grid cell (see MARGO Project Members, 2009).

10    To constrain the deep ocean as well as the surface, we additionally used data on the oxygen isotopic composition of calcite shells $(\delta^{18}\mathrm{O_c})$ of benthic and planktic foraminifera. The oxygen isotopic composition of seawater $(\delta^{18}\mathrm{O_{sw}})$ is locally linearly related to salinity and a conservative tracer of water masses. Its signal is preserved in the calcite shells of planktic and benthic foraminifera. To obtain a data coverage as high as possible, we collected data synthesis' from different sources, namely those of Marchal and Curry (2008) for benthic data, Waelbroeck et al. (2014) for planktic data, and of Caley et al. (2014), which contains benthic and planktic data. We additionally added the benthic $\delta^{18}\mathrm{O_c}$ data used in Völpel et al. (2018), including new data from 3 cores, and a new data compilation of LGM $\delta^{18}\mathrm{O_c}$ data, which contains published data from various sources and one unpublished data point. The new data compilation is available in the supplementary material. LGM-LH anomalies are given in the data sets of Waelbroeck et al. (2014), Caley et al. (2014) and Völpel et al. (2018). Distinct data sets for the LGM and the Late Holocene are provided by Marchal and Curry (2008). To compute anomalies from the LGM data of the new compilation, we used the Late Holocene data collection of Waelbroeck et al. (2005). We computed anomalies for the data set of Marchal and Curry (2008) and the new data compilation when data for both time slices from the same core and from the same foraminiferal species were available and excluded all other data points.

    Spatially varying uncertainty estimates are available for the data sets of Caley et al. (2014), Waelbroeck et al. (2014), Völpel et al. (2018) and the new data compilation. A spatially uniform uncertainty estimate of $0.18\,‰$ for each time slice is estimated by Marchal and Curry (2008). For a few data points from the synthesis of Waelbroeck et al. (2014), Caley et al. (2014) and the new data compilation no uncertainty estimates were available. In these cases, an uncertainty of $0.18\,‰$ was assigned for each time slice. We combined uncertainties for the two time slices (LGM and LH) according to the propagation of uncertainties via $\sigma_{\mathrm{LGM-LH}} = \sqrt{\sigma_{\mathrm{LGM}}^2 + \sigma_{\mathrm{LH}}^2}$. An overview of the proxy data that were used in this study can be found in Table 3 and their locations are shown in Fig. 2.

30    Similarly to the SST data, we averaged the $\delta^{18}\mathrm{O_c}$ data onto the model grid. The planktic $\delta^{18}\mathrm{O_c}$ were assigned to the surface layer of the model grid covering 0–50 m. To assign the benthic $\delta^{18}\mathrm{O_c}$ data to a model grid level, we subtracted $130\,\mathrm{m}$ from the core depths to account for the mean sea level change (Clark et al., 2009), and assigned the data to the respective vertical level. As in some cases multiple data points fell into the same grid cell, the weighted mean was computed according to the respective uncertainties resulting in 200 grid cells with benthic and 136 grid cells with planktic data. As proposed by the MARGO Project



Members (2009) and applied for the SST data, we computed the uncertainty estimate for each grid cell as the sum of the weighted mean of the uncertainty estimates and the variance within one grid cell.

## 3 Results

### 3.1 Pseudo-proxy Experiments

In this section, the results from the pseudo-proxy experiments are presented. For the experiments that used only a part of the global data (P1-Atlantic, P1-Pacific, P2-Atlantic, P2-Pacific), the cost function using only the respective data as well as the cost function including all data are analyzed. The cost function including all data is referred to as the global cost function, whereas the cost function including only the data in the respective part of the ocean is referred to as basin-specific cost function.

In the pseudo-proxy experiments, we use the Atlantic Meridional Overturning Circulation (AMOC) strength as a measure

for the success in reconstructing the ocean circulation. The AMOC is an important component of the global ocean circulation and has been the target in many paleoceanographic studies (e.g., Lynch-Stieglitz et al., 2007; Kurahashi-Nakamura et al., 2014; Muglia and Schmittner, 2015; Lippold et al., 2016). The maximum AMOC strength always refers to the maximum in the center of the upper AMOC streamfunction cell.

### 3.1.1 Set 1

In the first set of experiments the strong-constraint assumption held true and differences between the target run and the first guess arise only from different values of the control variables. The first guess has a maximum AMOC strength of 17.2 Sv compared to 12.3 Sv in the target and it has especially high differences in the simulated oxygen isotopic composition (Table 4). The FDS-IKS was very successful in reconstructing the target circulation and in reducing the cost function in the experiments using all data (P1-All) and using only the Atlantic data (P1-Atlantic, Fig. 3). Table 4 shows that the reconstructed circulation

in P1-All agrees perfectly with all three pseudo-proxy data types (SST, surface and deep ocean $\delta^{18}O_{sw}$) and that the maximum strength of the AMOC was faithfully reconstructed. All control variables were adjusted towards the correct target values in this experiment except for the high-to-low latitudinal gradient in the isotopic composition of precipitation (control variable 12). However, not all agree with the target values within the estimated uncertainty (Table 2).

Similarly promising are the results in P1-Atlantic. The AMOC strength was well reconstructed and the cost function indicates

an almost perfect agreement between model and data for this experiment (Fig. 3 and Table 4). The estimated control variable values are similarly good but three control variables were adjusted in the wrong direction (control variables 5, 7, and 12, Table 2). In the experiment that used only the Pacific and Indian Ocean data (P1-Pacific) the cost function shows a large decrease in the first iteration indicating a perfect agreement of the model with the basin-specific data (Table 4) and small mismatches in the global cost function. The control variable values were not well estimated with only very few control variables agreeing with

the target values within the estimated uncertainty and four control variables that were adjusted in the wrong direction (control variable 9, 10, 12, and 15). In the following iterations the overturning collapsed due to the changes in the air temperature and





precipitation field resulting in a strong increase of the cost function. The basin-specific cost function was again reduced and shows a perfect agreement of model and data in iterations 5 to 10 in which the AMOC is collapsed (solid line in Fig. 3). The global cost function, however, indicates a large model-data mismatch for the Atlantic data in these iterations (dashed line in Fig. 3 and numbers in brackets in Table 4).

In all experiments we observed a drastic reduction of the cost function in the first iteration and then a further slight decrease or increase in the following iterations. A large increase in the cost function from one iteration to the next is in all cases related to a change from an active Atlantic overturning to a collapsed overturning or vice versa (Fig. 3, 4, and 7). An exception is iteration 2 in P2-Pacific, where the large increase of the cost function was caused by the activation of a vigorous Pacific overturning. The collapse or activation of the Atlantic or Pacific overturning reflects a strongly non-linear reaction of the model to changes

in the control variables. The FDS-IKS, which is based on a local quadratic approximation of the cost function, cannot predict this behavior. The collapse of the overturning circulation is related to changes in the air temperature and precipitation fields, possibly an increase of precipitation in the Atlantic compared to the previous iteration in P1-All, iteration 10, P1-Atlantic, iteration 9, and P1-Pacific, iteration 2, or by an interaction of the changes in air temperature and precipitation.

### 3.1.2   Set 2

Target 2 was generated by perturbing eight control variables in the isotopic composition of precipitation and water vapor and by additionally applying a freshwater hosing in the North Atlantic. The freshwater hosing was not applied in the first guess and it was not estimated during the optimization processes. The missing hosing emulates an unknown model error or bias that is not accounted for in the optimization, and hence, the strong-constraint assumption did not hold true in these experiments. The target has an AMOC strength of 13.6 Sv compared to 17.2 Sv in the first guess and it has especially high differences to the first

guess in the oxygen isotopic composition (Table 4, Fig. 5 and 6).

In the pseudo-proxy experiment reconstructing Target 2 using all data (P2-All) the normalized cost function was reduced to a value of one indicating a perfect model-data agreement and the AMOC strength was very well reconstructed (Fig. 4 and Table 4). The control variable values, however, were not well estimated and only a few agree with the target valuess within their uncertainties (Table 2). Nevertheless, the target–reconstruction misfit in Fig. 5 and 6 show very small differences at the surface

as well as in the deep ocean in the Atlantic, Pacific and Indian Ocean. Only in the Arctic Ocean, where no data is available, the reconstruction has much lower values.

The cost function in the experiment using only the Atlantic data (P2-Atlantic) was successfully reduced in the first iteration. It indicates a perfect model-data agreement for the benthic $\delta^{18}O_c$ data and an almost perfect agreement for planktic $\delta^{18}O_c$ and SST data (Table 4). In the later iterations the basin-specific cost function is still low (solid line in Fig. 4), the global

cost, however, indicates a large discrepancy for the data outside of the Atlantic Ocean (dashed line in Fig. 4). The AMOC was adjusted towards the target AMOC strength but remains too strong in all iterations. The surface and deep ocean target–reconstruction differences are similar to those in P2-All, but show bigger difference at the surface in the Pacific and Indian Ocean as well as in the Atlantic Ocean. The surface and subsurface waters in the northern Atlantic and Pacific subtropical gyre





are slightly less enriched than in the target (Fig. 5 and 6). As in P2-All the control variable values were not well estimated and only a few agree with the target values within the estimated uncertainty.

The experiment using only the Pacific data (P2-Pacific) shows a good reduction of the cost function in the first iteration. The basin-specific cost function stayed close to one during the optimization except in iteration 2, where a Pacific overturning was

activated (not shown). The AMOC collapsed in the first iteration and the global cost function indicates large discrepancies for the Atlantic data in all iterations except the last, where the AMOC resumed (Fig. 4). The collapse and activation of the Atlantic and Pacific overturning were caused by changes in the air temperature and precipitation fields. The control variable values, for example, show a decrease of freshwater input through precipitation and a decrease of air temperature from iteration 1 to 2 over the Pacific. These changes or their interaction must have caused the activation of the Pacific overturning. The surface

target–reconstruction misfit in the Pacific is very small even if the AMOC is collapsed (Fig. 5 and 6), but in the Atlantic the surface as well as the deep ocean show a large mismatch in these iterations. The missing North Atlantic Deep Water is clearly visible in Fig. 6.

## 3.2   LGM Experiment

The LGM first guess simulation has a maximum AMOC strength of 17.2 Sv (Fig. 7). The simulated SST field is in very good

agreement with the MARGO SST reconstruction, but especially the simulated surface $\delta^{18}O_c$ anomalies show a large mismatch to the planktic $\delta^{18}O_c$ anomalies data (Table 4 and Fig. 7). The data show a much stronger gradient between high and low latitudes with much higher values in the low latitudes than are found in the model (Fig. 8).

During the optimization the misfit cost function ($J_{\text{misfit}}$) was greatly reduced. The largest reduction of the misfit cost function was reached in iteration 3. Especially the simulated surface $\delta^{18}O_c$ shows a large improvement. The cost function measuring

the misfit to the planktic and benthic $\delta^{18}O_c$ data was reduced by 77.6 % and 30.8 % in the third iteration, respectively. The SST misfit increased very slightly, but the model still shows a good agreement with the SST data after the optimization (Table 4, Fig. 7). The too weak gradient in simulated $\delta^{18}O_c$ anomalies was much improved, but major discrepancies remain between the simulated surface $\delta^{18}O_c$ anomalies and the planktic $\delta^{18}O_c$ anomalies data. The mismatch is especially high in the upwelling regions close to the African and North American coast, and in the north-west Atlantic Ocean (Fig. 8). Almost all of the benthic

data in the deep Atlantic Ocean show agreement with the model. Only in the higher levels above 1,000 m depth the majority of data points indicate a mismatch (Fig. 9). In iteration 3 the AMOC was reduced to a maximum strength of 13.8 Sv (Fig. 10) and the North Atlantic Deep Water (NADW) is about 700 m shallower than in the first guess reaching until approximately 2,500 m depth.

The improvement in the model-data misfit was achieved by substantial changes in the control variables (Table 2 and 4,

and Fig. 1). While the changes in the air temperature and precipitation stayed within one or two times the assumed standard deviation, the changes in the isotopic composition and water vapor and precipitation greatly exceed this limit.

In iteration 4 of the optimization the AMOC collapsed likely due to warmer air temperatures over the North Atlantic Ocean produced by a higher value in the mean Atlantic air temperature (control variable 1) and a decrease in the high-to-low latitudinal gradient of air temperature over the Atlantic Ocean (control variable 3, not shown). The collapse of the AMOC resulted in a

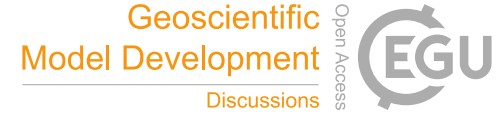

strong increase of the cost function. In the following iterations the AMOC stayed inactive, the cost function, however, again decreased (Fig. 7). In iteration 7 the cost function shows similarly low values than in iteration 3 with only slightly higher values in the benthic and planktic $\delta^{18}O_c$ data (Table 4 and Fig. 7) even though the AMOC is inactive.

## 4   Discussion

### 4.1   Data Assimilation Approach

The pseudo-proxy experiments showed that the FDS-IKS in combination with the state reduction approach is capable of efficiently reducing the cost function and reconstructing the global ocean circulation from SST and $\delta^{18}O_{sw}$ data at a realistic LGM data coverage and with realistic LGM uncertainties. The major part of the cost function reduction typically occurred in the first three iterations. Each iteration requires $N \cdot p + 1$ ensemble members where $N$ is the number of perturbation experiments per control variable and $p$ the number of control variables. In our experiments, $3 \cdot 16 + 1 = 49$ ensemble members were run in parallel for each iteration to estimate 16 control variables. We performed 10 iterations for each experiment. With more expensive models, however, it might be sufficient to only perform about 3 iterations to achieve a substantial reduction of the cost function or, to further reduce the computational cost, the number of control variables can be reduced. The method is, therefore, extremely fast and requires less computational time than previous methods for parameter estimation (e.g., Annan et al., 2005b; Breitkreuz et al., 2018).

Even though the method is able to successfully reconstruct the ocean circulation when enough data is available, not all control variable values are accurately estimated (Table 2). We partly attribute the errors in the estimated control variable values to the FDS-IKS and the assumptions that come with it, and partly to an insufficient constraint by the pseudo-proxy data at LGM data coverage (see next section). Several assumptions are made in the derivation of the FDS-IKS. First, the observational errors and errors in the first guess are assumed to be Gaussian. This assumption is made for most other data assimilation methods, even though it is hardly ever true in real applications. In the application here the uncertainty distribution of the LGM data resembled a lognormal distribution and the uncertainties for the pseudo-proxy experiments were, therefore, generated in this way. Breaking the Gaussian assumption, however, did not seem to have a large negative influence on reconstructing the circulation. Second, the FDS-IKS locally approximates the cost function and reduces it iteratively starting from a first guess. As with the adjoint method (Wunsch, 1996), the final result depends strongly on the first guess and it is possible that the estimated control variables reflect a local minimum of the global cost function. Third, non-linear effects originating from correlations between the control variables are not taken into account in the sensitivity matrix $\mathbf{G}$ due to the one-at-a-time sensitivity approach. Each column of $\mathbf{G}$ holds the sensitivity of the model in observation space to one control variable with the other control variables fixed at the current best guess values. This limitation is specific to the one-at-a-time approach of the FDS-IKS. It could be avoided with an ensemble approach where the control variables for each ensemble member vary in all control dimensions. An ensemble approach, however, can suffer from spurious correlations or demand a much bigger ensemble size (Evensen, 2009). Additionally, we assumed the observational error covariance matrix and the background error covariance matrix to be diagonal. This assumption is as well made in many other data assimilation applications because values





for correlations are hard to estimate. As the off-diagonal entries are zero, correlations between observational uncertainties and between control variable uncertainties are assumed to be zero. Note, that the updated error covariance matrix for the optimized control variables (Eq. 8) is not diagonal and provides an estimate of the correlations.

The control variables in Set 2 are estimated less accurately than those in Set 1 (Table 2). The reason for the inaccurately estimated control variables in Set 2 is that the strong-constraint assumption does not hold true in these experiments and the control variables try to account for the additional model error. The first guess of the control variables for air temperature and precipitation (control variables 1-8) are zero and coincide with the target values, but they were adjusted during the optimization to account for the missing freshwater hosing. The control variables for the isotopic composition of precipitation and water vapor (control variables 9-16) seem to be influenced by the different control variable values for air temperature and precipitation and the missing freshwater hosing as well.

The disregard of the non-linear effects originating from correlated control variables due to the FDS-IKS approach might be one reason why the method does not seem to converge properly (Fig. 3 and 4). As the sensitivity of the model with respect to one control variable is computed based on the current best guess of the other control variables, the sensitivity might change in the next iteration when the best guess of the other control variables is changed. This can result in back-and-forth changes of the control variables. Another reason for the seemingly missing convergence is the disruption of the optimization due to the AMOC collapse. The method is based on a locally linear approximation of the cost function and cannot predict this non-linear jump in the cost function. The reason for the AMOC collapse might either be unrealistically large changes in the atmospheric forcing or it might be that the model is close to an unstable state. Extreme values for the control variables can be restricted by smaller first guess uncertainties, but the first guess uncertainties are hardly ever known in applications and are usually chosen in an ad-hoc fashion.

## 4.2 LGM Data Coverage Constraint

The ocean state is well reconstructed in the pseudo-proxy experiments using all data (P1-All, P2-All) and using only the Atlantic data when the strong-constraint assumption is fulfilled (P1-Atlantic). The cost functions in P1-All, P2-All, and P1-Atlantic stay almost constant close to 1.0 for all iterations (except for iterations 10 in P1-All and 9 in P1-Atlantic where the AMOC collapses). The AMOC varies in a range of approximately $2\,\mathrm{Sv}$ around the target strength in these iterations (Fig. 3 and 4). These results indicate that in a perfect model set-up the data at LGM coverage (P1-All) and even only the Atlantic data at LGM coverage (P1-Atlantic) are sufficient to constrain the ocean circulation and the AMOC within a range of a few Sverdrups. Even when the strong-constraint assumption does not hold true, the global data at LGM coverage are sufficient to constrain the AMOC strength (P2-All). In all iterations with a low cost function in these experiments the vertical extent of the southward transport and the extent of deep water masses is very similar. For example, in experiment P2-All iterations 1 and 8 have a maximum AMOC strength of $16.3\,\mathrm{Sv}$ and $14.1\,\mathrm{Sv}$, respectively, while the vertical extent of the southward transport is almost the same reaching depths of $2,000\,\mathrm{m}$ and $2,100\,\mathrm{m}$ at $30°\,\mathrm{S}$ (not shown). This similarity might explain why a variation of a few Sverdrups in maximum AMOC strength has no effect on the cost function and why the AMOC strength is not constrained more accurately by the data.





Similarly, in iteration 1 of P2-Atlantic the cost function is very close to one, but the AMOC is stronger (16.6 Sv) than in the target (13.6 Sv). Figure 6 shows that the remaining target–reconstruction $\delta^{18}O_{sw}$ differences are small in the deep ocean and larger closer to the surface in P2-Atlantic. The signal of the less enriched surface and subsurface waters is transporter by a stronger AMOC. The two effects might balance each other, which could explain the perfect model data misfit in the deep

ocean in P2-Atlantic (Table 4) while the AMOC is too strong.

The reason why the surface conditions and therefore the AMOC are not as accurately reconstructed as in P2-All might be related to the missing Pacific and Indian Ocean data. In experiment P2-Atlantic the surface model-data misfit in the Atlantic Ocean is not smaller than that in the Pacific even though Atlantic data is used but no Pacific data (Fig. 5). This indicates, first, that the Atlantic data places a significant constraint on the Pacific and Indian Ocean, and second, because the surface Atlantic

Ocean is not as accurately reconstructed as in P2-All, that the constraint of data from the Pacific and Indian Ocean is needed to get an accurate representation of the surface ocean globally. The accurate representation of the surface conditions in the Atlantic Ocean in turn influences how well the AMOC is reconstructed. An explanation for this might be that the influence of a wrong surface signal in the Indian Ocean could be transported through, for example, the Agulhas and the Benguela Current to the Atlantic Ocean. Another explanation of how the Pacific data influence the Atlantic Ocean is through the Antarctic Circumpolar

Current and the formation of Antarctic Bottom Water. However, only very small differences in the deep ocean properties in the Pacific and the Atlantic Ocean between P2-All and P2-Atlantic could be found. This does not mean that this connection does not exist, but merely that in our experiments it does not show. The basin-specific cost functions in the experiments using only the Pacific data (P1-Pacific, P2-Pacific) are very low even in iterations where the AMOC is collapsed. Unsurprisingly, only the Pacific data is not sufficient to constrain the AMOC strength.

To summarize, these results indicate that while the cost function reduction is similar in P2-All and P2-Atlantic, a very good reconstruction of the AMOC strength is aided by proxy data from the Indian and the Pacific Ocean. At the same time, the proxy data from the Atlantic Ocean is essential for the reconstruction of the AMOC and the AMOC is not at all constraint by data only from the Indian and Pacific Ocean.

Even though the circulation is well reconstructed and the cost function successfully reduced in experiments P1-All, P1-

Atlantic, P2-All, and P2-Atlantic, the control variable values are not estimated accurately. The target–reconstruction $\delta^{18}O_{sw}$ surface fields show large discrepancies in these reconstructions in the Arctic Ocean in P2-All and P2-Atlantic (Fig. 5). The inaccurate representation of the conditions in the Arctic Ocean is not present in temperature or salinity fields (not shown) and the estimated control variables for the isotopic composition of precipitation and water vapor correspond to too low values in the high-latitudes compared to the target values. It follows that the mismatch in the Arctic is due to inaccurately estimated control

variable values in the isotopic composition of precipitation and water vapor. For the Arctic Ocean no data is available except in the northern North Atlantic Ocean and more data would likely be necessary to accurately constrain these control variables.

### 4.3   LGM Ocean State Estimate

The optimization greatly reduces the misfit between the simulated LGM ocean state and the LGM proxy data in the first three iterations. Especially the simulated $\delta^{18}O_c$ shows a large improvement but major discrepancies remain after the optimization.



While the AMOC has a maximum strength of 13.8 Sv in iteration 3 (Fig. 10), it is inactive in iteration 7 of the optimization but the model-data misfit is similarly low as in iteration 3 (Table 4 and Fig. 7). The pseudo-proxy data experiments indicated that the LGM data coverage is sufficient to constrain the AMOC. The major difference between the pseudo-proxy and the LGM experiments is the remaining model-data misfit. The pseudo-proxy reconstructions reach a perfect fit to the pseudo-proxy data,

which provides a good AMOC constraint, but in the experiment with the reconstructed LGM data major model-data misfits remain and the AMOC is not well constrained in this case.

A recurring hypothesis in paleoceanographic research is that of an LGM Atlantic Ocean with a shallower NADW accompanied by a weaker, but nevertheless active, overturning circulation compared to the modern ocean (Lynch-Stieglitz et al., 2007; Muglia et al., 2018). The optimized simulation of iteration 3 concurs with this hypothesis. The NADW is shallower than in our

first guess and much shallower than in the optimized modern simulation of Breitkreuz et al. (2018) (Fig. 9 and their Fig. 4) and it has a weaker AMOC of 13.8 Sv compared to 16.6 Sv inferred for the modern ocean by Breitkreuz et al. (2018) (Fig. 10). Kurahashi-Nakamura et al. (2017) estimated the state of the LGM ocean with the adjoint method constrained by global SST and benthic $\delta^{18}O_c$ and $\delta^{13}C$ data from the Atlantic Ocean. They found a similarly shallow NADW but stronger AMOC of 21.3 Sv (Kurahashi-Nakamura et al., 2017). The difference in AMOC strength might be related to the missing constraint of

global planktic $\delta^{18}O_c$ data and/or benthic $\delta^{18}O_c$ data from the Indian and Pacific Ocean, or it can be related to the lack of carbon isotope data as constraint or the insufficiently reduced model-data misfit in our estimate. The remaining misfit to the benthic $\delta^{18}O_c$ data is highest above 1,000 m depth in our estimate (Fig. 9). The AMOC transports the surface $\delta^{18}O_{sw}$ signal to this area and the misfit could point to an incorrectly simulated AMOC strength and/or to incorrectly simulated surface conditions, which coincides with the remaining misfit to the planktic $\delta^{18}O_c$ data in the North Atlantic (Fig. 8). It follows that the

estimated AMOC strength needs to be interpreted very carefully as it seems to depend on many variables in the optimization process.

The improvement in our optimization is achieved by large changes in the control variables, which are on average bigger than the assumed standard deviation of the control variables (Table 2 and Fig. 1). The air temperature and precipitation, which influence the simulated temperature as well as simulated $\delta^{18}O_c$, show changes close to the assumed standard deviation.

The adjustments in the control variables in the isotopic composition of water vapor and precipitation, which only influence the simulated $\delta^{18}O_c$, exceed the limit of two standard deviations. Despite these high changes in the control variables, major model-data misfits remain in the optimized LGM simulation. The successful pseudo-proxy experiments indicate that the remaining model-data misfit in the LGM experiment is not due to insufficiencies of the FDS-IKS but rather that the model inhabits large errors that are not related to the control variables. The strong-constraint assumption does not hold true in this experiment.

While this is also true for the pseudo-proxy experiments of Set 2, the unknown model errors that are not accounted for in the LGM optimization are likely much bigger than those generated through the missing freshwater hosing in the pseudo-proxy experiments. The changes in the control variables try to account for these model errors and therefore, they need to be interpreted very carefully. To avoid this in future work, additional errors can be added to the assumed uncertainties of the observations (e.g., Kurahashi-Nakamura et al., 2017; Breitkreuz et al., 2018), but they are hard to determine and might provide too much

freedom in the optimization. Another reason for the remaining model-data misfit might be that the observational data include



errors that are not accounted for in the assumed uncertainties. Assuming higher errors in the cost function would reduce the cost function value and demand smaller changes in the control variables.

In fact, the remaining model-data misfit cannot be improved by our control variables. The remaining $\delta^{18}O_c$ anomalies model-data misfit is highest in the western North Atlantic Ocean (Fig. 8). Whereas simulated values around $40°\,N$ are too high and the

values around $80°\,N$ are too low, the values around the equator show a good agreement. These differences cannot be reduced by the control variables as only the mean, the north-to-south gradient and the high-to-low latitudinal gradient in the forcing fields are controlled. Estimating the coefficients of higher degree Legendre polynomials might improve these differences. However, the data in the eastern part of the Atlantic Ocean generally show a good agreement, and might inhibit further adjustments of the isotopic forcing fields. In other words, the remaining model-data misfit shows a complex horizontal structure that cannot

be mirrored by the control variables that control large-scale latitudinal patterns.

The remaining model-data misfit might be due to errors in the atmospheric forcing with other spatial patterns or due to various other internal model uncertainties like the parameterization of mixing in the ocean or the low model resolution. The error patterns in the western North Atlantic Ocean hint at an incorrect representation of the Gulf Stream in the model. The Gulf Stream is typically not well represented in ocean models with low resolutions (Drews et al., 2015). Some $\delta^{18}O_c$ anomalies

model-data misfit remains in the upwelling areas close to the western coasts of southern Africa and North America and in the eastern equatorial Pacific. The upwelling is not well represented in the model due to the low grid resolution. The model-data misfit in the Southern Ocean slightly increased during the optimization. For the isotopic composition of water vapor and precipitation the mean and high-to-low latitudinal gradients are estimated. The adjustment of the high-to-low latitudinal gradients is balanced in iteration 3 between the too low values in the northern high latitudes and the too high values in the

southern high latitudes (Fig. 8). This again indicates that other model errors are causing the remaining model-data misfit.

To further improve the LGM estimate, more control variables would need to be optimized. It would, however, increase the computational demand and it is not easy to determine which parameters would have the desired effect. One possibility is to include higher order terms of the Legendre polynomials, but it would likely not have a large effect on east-west differences within one ocean basin. Other possibilities are to separate the Atlantic Ocean in an eastern and in a western part and alter the

atmospheric forcing for each of the parts separately, but this approach might result in unphysical borders between the ocean basin parts. Additionally, as non-linear effects from correlated control variables are not taken into account, an introduction of more control variables in one forcing field might result in less accurate results. Another possibility would be to include the estimation of the wind fields, the vertical or horizontal diffusion coefficients or other internal model parameters. An ocean model could also be coupled to a (possibly simplified) atmospheric model and parameters in the atmospheric model that affect

the air-sea fluxes could be estimated (e.g., García-Pintado and Paul, 2018). The FDS-IKS can easily be applied with other ocean and climate models.

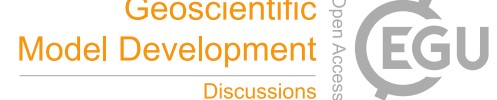

## 5   Conclusions

We presented a new technique for ocean state estimation and for estimating uncertain spatially varying boundary conditions that combines the FDS-IKS with a state reduction approach. The technique has shown to be very efficient and to require substantially less computing time than other methods. It is very successful in reducing the cost function and reconstructing the ocean circulation when data at LGM coverage is available and when other model errors are not too big.

We used the method to investigate the data constraint of proxy data outside of the Atlantic Ocean on the Atlantic overturning circulation. Our results indicate that while data from the Indian and Pacific Ocean is required to accurately reconstruct the global upper ocean and therefore the AMOC strength, data from the Atlantic Ocean is far more important.

We applied the method to estimate the global ocean state during the LGM. The method greatly reduces the model-data misfit, but some model-data misfit remains for the planktic and benthic $\delta^{18}O_c$ data. The remaining model-data misfit is likely due to additional model errors and it is not able to be reduced further with the current control variables. The method produces two ocean state estimates with a similar model-data misfit, one with an active AMOC and one with a collapsed circulation. With the remaining model-data misfit, the available proxy data is not able to successfully constrain the LGM ocean circulation. The reconstructed active LGM ocean state, however, is consistent with the results of previous studies and shows a weaker AMOC and a shallower NADW than the modern ocean.

*Code and data availability.*   The authors are currently submitting the new $\delta^{18}O_c$ data compilation to the data base PANGAEA (https://www.pangaea.de/). The code for the state reduction approach and the FDS-IKS is available at GitHub via https://github.com/cbreitkreuz/FDS-IKS-and-state-reduction-approach. The MITgcm model code is publicly available via http://mitgcm.org/.

*Author contributions.*   A. Paul, M. Schulz and S. Mulitza were involved in the funding acquisition and supervision of the project. A. Paul, M. Schulz and C. Breitkreuz designed the experiments and C. Breitkreuz carried them out. J. García-Pintado initiated the use of the FDS-IKS, first developed it and assistet in the implementation. C. Breitkreuz implemented the data assimilation method and state reduction approach for the MITgcm and created all figures and tables in the manuscript. S. Mulitza and A. Paul collected the new LGM $\delta^{18}O_c$ data compilation. All versions of the manuscript were written by C. Breitkreuz. All coauthors reviewed and commented on different versions of the manuscript.

*Competing interests.*   The authors declare that they have no conflict of interest.

*Acknowledgements.*   This study was supported by the German Federal Ministry of Education and Research (BMBF) as a Research for Sustainability initiative (FONA) through the German Climate Modeling Initiative PalMod project (FKZ: 01LP1511D). We are grateful to the MARGO Project Members, to Claire Waelbroeck and coauthors, to Thibaut Caley and coauthors, and to Olivier Marchal and William





Curry for synthesizing the LGM data and making them publicly available, and to Rike Völpel for providing the water isotope package for the MITgcm.



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





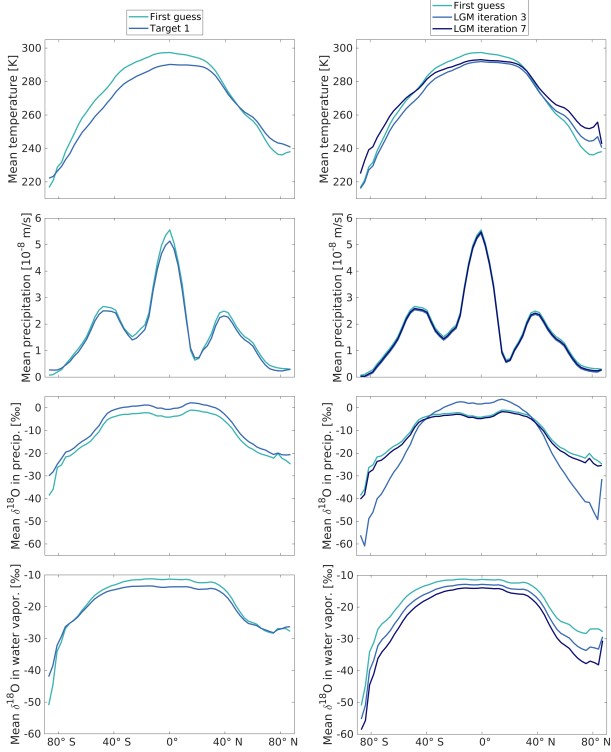

**Figure 1.** Meridional mean of air temperature, precipitation, isotopic composition of precipitation and of water vapor over the Atlantic in the first guess compared to Target 1 (left side) and in the first guess compared to the optimized LGM estimate in iteration 3 and 7 (right side). Note that the air temperature and precipitation are not altered for Target 2 and hence, they coincide with the first guess, and that the isotopic composition of precipitation and of water vapor in Target 2 coincide with those in Target 1.





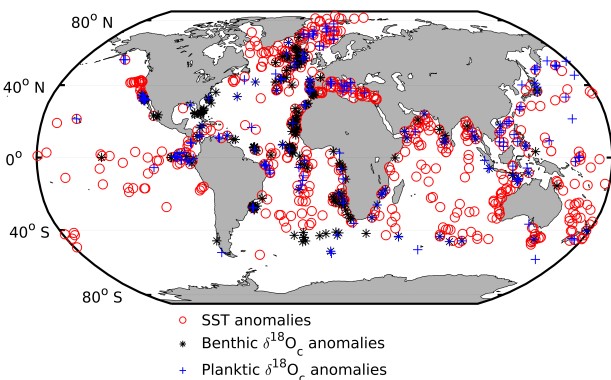

**Figure 2.** Locations of the LGM benthic and planktic $\delta^{18}O_c$ data as well as the SST data used in the LGM optimization.





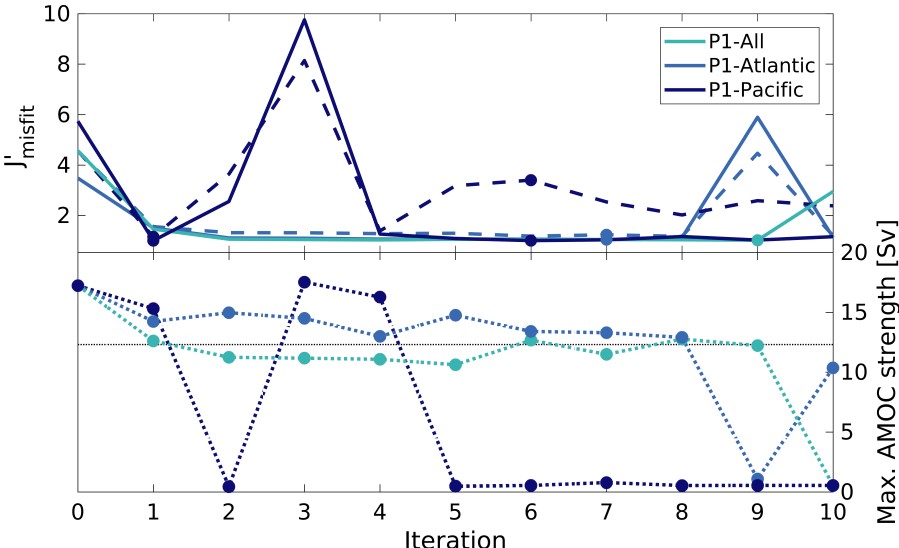

**Figure 3.** Normalized misfit cost function $J'_{\mathrm{misfit}}$ (upper panel) and max. AMOC strength (lower panel) over the iterations of the optimization process for experiments P1-All (using all data), P1-Atlantic (using only data from the Atlantic), and P1-Pacific (using only data from outside of the Atlantic). The normalized misfit cost function $J'_{\mathrm{misfit}}$ is the model-data misfit cost function $J_{\mathrm{misfit}}$ divided by the number of model-data comparisons. The solid lines show the basin-specific cost function including only the data that were used in the respective experiment. Dashed lines in the upper panel show the global cost function including all observations. The dots in the upper panel highlight the iterations that are given in Table 4. The thin horizontal line in the lower panel indicates the target maximum AMOC strength (Target 1).





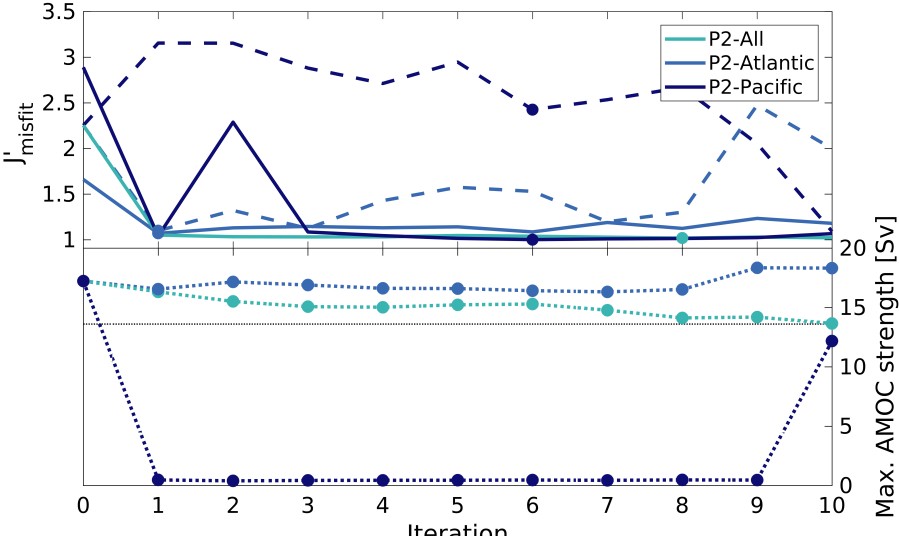

**Figure 4.** Normalized misfit cost function $J'_{\text{misfit}}$ (upper panel) and max. AMOC strength (lower panel) over the iterations of the optimization process for experiments P2-All (using all data), P2-Atlantic (using only data from the Atlantic), and P2-Pacific (using only data from outside of the Atlantic). The normalized misfit cost function $J'_{\text{misfit}}$ is the model-data misfit cost function $J_{\text{misfit}}$ divided by the number of model-data comparisons. The solid lines show the basin-specific cost function including only the data that were used in the respective experiment. Dashed lines in the upper panel show the global cost function including all observations. The dots in the upper panel highlight the iterations that are given in Table 4. The thin horizontal line in the lower panel indicates the target maximum AMOC strength (Target 2).





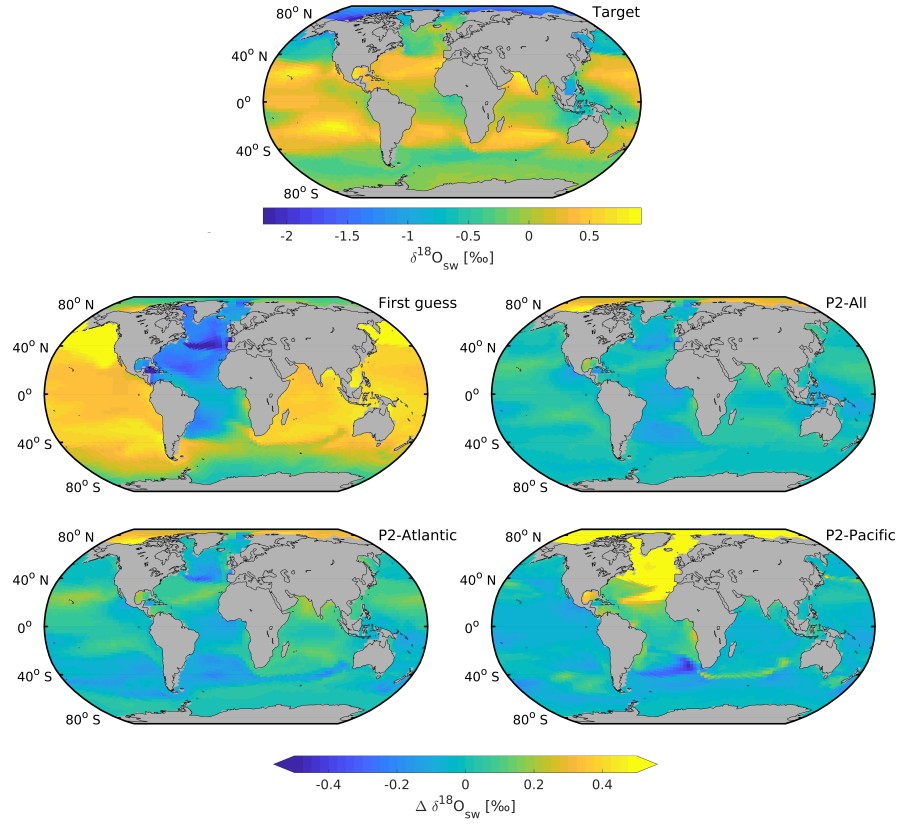

**Figure 5.** Simulated 100-year mean surface $\delta^{18}O_{sw}$ $(0-50\,\text{m})$ in Target 2 and target–reconstruction differences from the first guess, reconstructions P2-All, iteration 8, P2-Atlantic, iteration 1, and P2-Pacific, iteration 6.





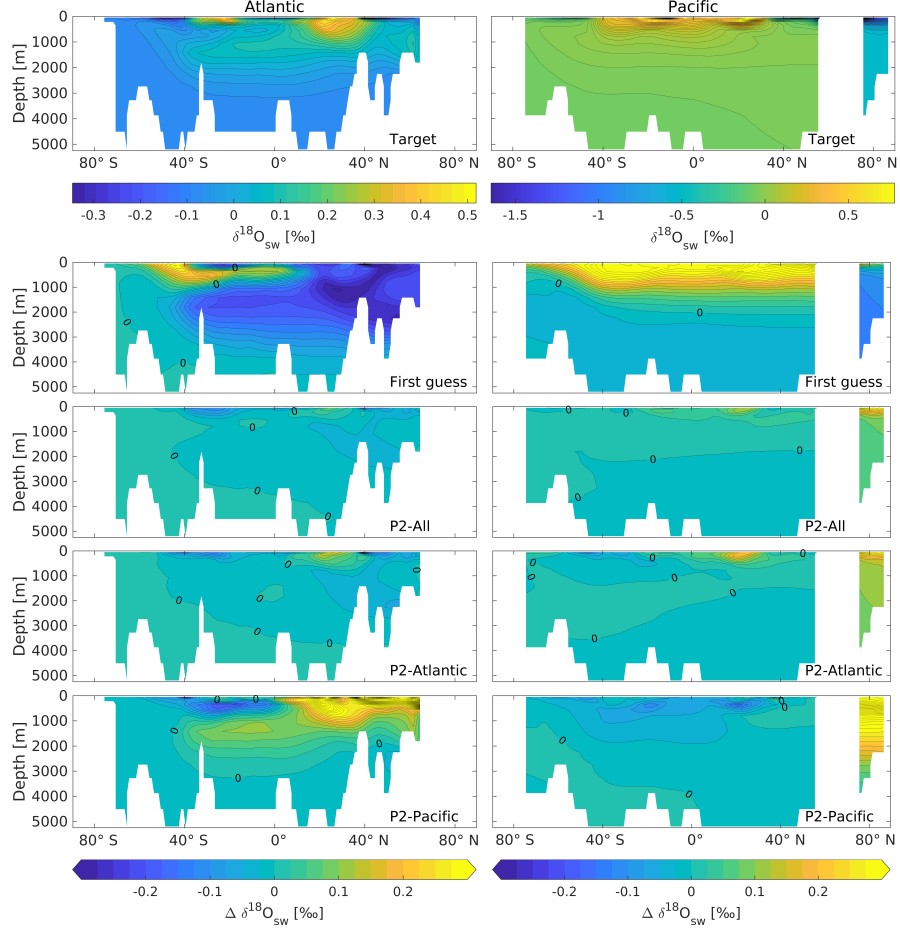

**Figure 6.** Vertical transects through the Atlantic Ocean at 32.5° W and the Pacific Ocean at 150° W of the simulated 100-year mean $\delta^{18}O_{sw}$ in Target 2 and target–reconstruction differences from the first guess, the reconstructions P2-All, iteration 8, P2-Atlantic, iteration 1, and P2-Pacific, iteration 6.





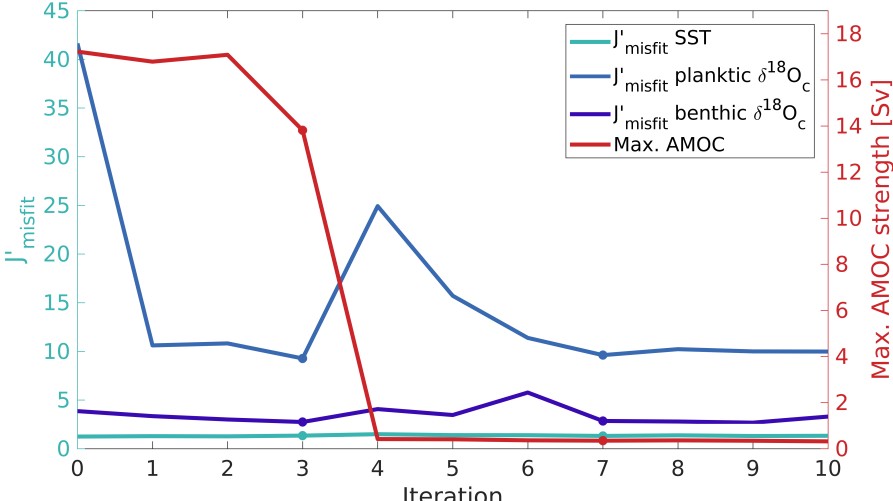

**Figure 7.** Normalized misfit cost function $J'_{\mathrm{misfit}}$ for different data types and max. AMOC strength over the iterations of the optimization process in the LGM experiment. The dots highlight the iterations that are analyzed further in the text and given in Table 4.

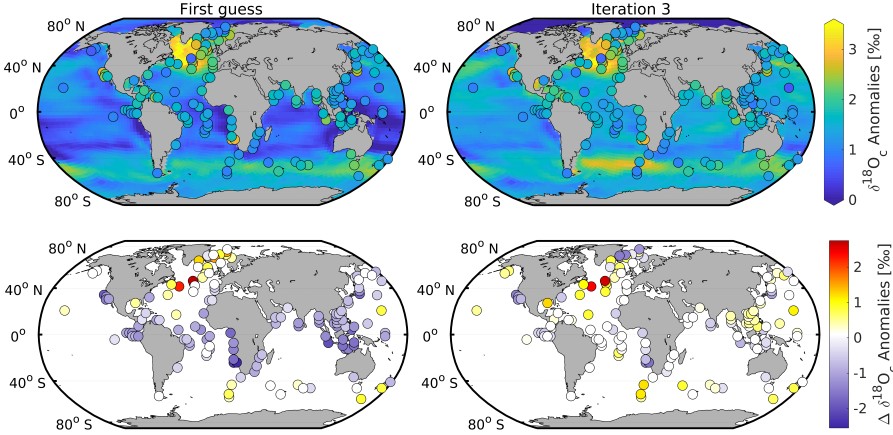

**Figure 8.** Simulated 100-year mean surface field (0–50 m) of $\delta^{18}O_c$ LGM-LH anomalies in the first guess and LGM optimization (iteration 3) and assimilated planktic $\delta^{18}O_c$ data (upper panels), and respective model-data differences (lower panels). Differences smaller than the uncertainty of the observational data are displayed in white.





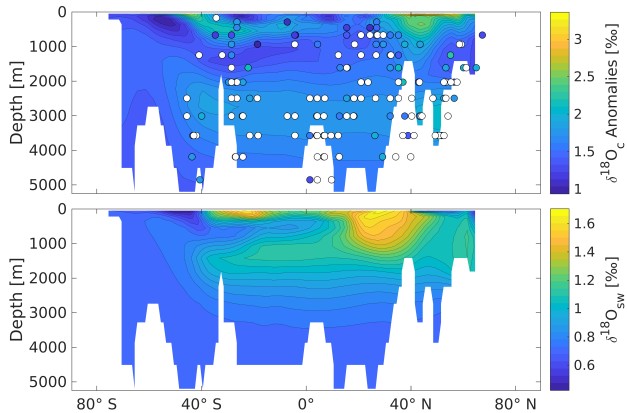

**Figure 9.** Simulated 100-year mean $\delta^{18}O_c$ LGM-LH anomalies (upper panel) and simulated 100-year mean $\delta^{18}O_{sw}$ at a vertical transect through the Atlantic Ocean at 32.5° W in the LGM optimization (iteration 3) and assimilated benthic $\delta^{18}O_c$ data. Data points where the model agrees with the observational data within their respective uncertainties are displayed in white.





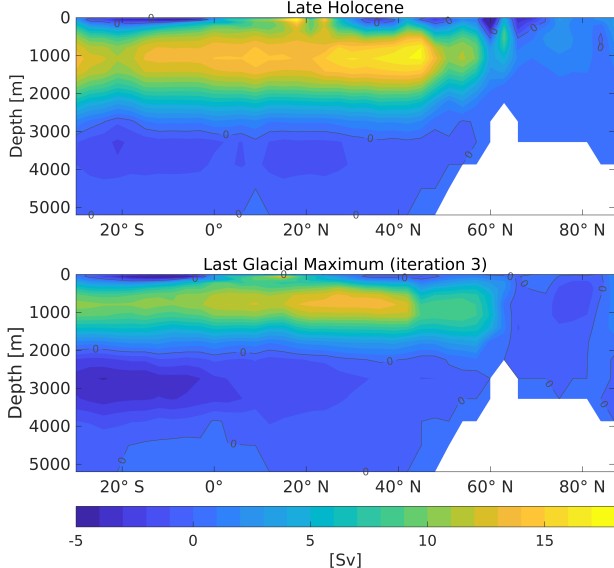

**Figure 10.** Simulated 100-year mean Atlantic Meridional Overturning Circulation streamfunction in the Late Holocene estimate of Breitkreuz et al. (2018) and in our LGM optimization (iteration 3).





**Table 1.** List of control variables. The part of the global ocean over which the control variable is estimated is indicated in the column "Area".

Note that the actual control variables, which are estimated in the optimization, are the corresponding Legendre polynomial coefficients.

|  | Field | Control variable | Area |
|---|---|---|---|
| 1 | **Air temperature** | Mean | Atlantic Ocean |
| 2 | | North-to-south gradient | Atlantic Ocean |
| 3 | | High-to-low latitudinal gradient | Atlantic Ocean |
| 4 | | Mean | Pacific and Indian Ocean |
| 5 | | North-to-south gradient | Pacific and Indian Ocean |
| 6 | | High-to-low latitudinal gradient | Pacific and Indian Ocean |
| 7 | **Precipitation** | Mean | Atlantic Ocean |
| 8 | | Mean | Pacific and Indian Ocean |
| 9 | **Isotopic composition** | Mean | Atlantic Ocean |
| 10 | **of precipitation** | High-to-low latitudinal gradient | Atlantic Ocean |
| 11 | | Mean | Pacific and Indian Ocean |
| 12 | | High-to-low latitudinal gradient | Pacific and Indian Ocean |
| 13 | **Isotopic composition** | Mean | Atlantic Ocean |
| 14 | **of water vapor** | High-to-low latitudinal gradient | Atlantic Ocean |
| 15 | | Mean | Pacific and Indian Ocean |
| 16 | | High-to-low latitudinal gradient | Pacific and Indian Ocean |





**Table 2.** List of control variable values in the target runs and estimated values in the pseudo-proxy experiments and in the LGM optimization. Resulting meridional means over the Atlantic for the target runs and the LGM simulations are shown in Figure 1. Note that Target 2 is additionally altered with a freshwater hosing of 0.24 Sv in the North Atlantic Ocean, which is not a control variable in the optimization. The iteration with the smallest model-data misfit is given for each of the pseudo-proxy experiments and the LGM optimization, except for the additional iteration for the LGM experiment (iteration 7). The assumed standard deviations for the first guess control variables are chosen in an ad-hoc fashion according to the influence of the control variable on the meridional mean of the forcing field. Estimated standard deviations (*Std.*) for the optimized control variables are given in italics below each estimated control variable value. Note that the first guess simulation has values of 0 for all control variables. Control variables 1-16 refer to the control variables in Table 1. *(Table is continued on next page.)*

| | Control variable | | | | | | | |
| --- | --- | --- | --- | --- | --- | --- | --- | --- |
| | 1 | 2 | 3 | 4 | 5 | 6 | 7 | 8 |
| Assumed std. | $5 \cdot 10^0$ | $5 \cdot 10^0$ | $5 \cdot 10^0$ | $5 \cdot 10^0$ | $5 \cdot 10^0$ | $5 \cdot 10^0$ | $1 \cdot 10^{-9}$ | $1 \cdot 10^{-9}$ |
| **Target 1** | $-5.0 \cdot 10^0$ | $9.0 \cdot 10^0$ | $5.0 \cdot 10^0$ | $-1.7 \cdot 10^0$ | $-4.3 \cdot 10^0$ | $0.2 \cdot 10^0$ | $-1.6 \cdot 10^{-9}$ | $1.7 \cdot 10^{-9}$ |
| P1-All (iter. 9) | $-3.4 \cdot 10^0$ | $9.1 \cdot 10^0$ | $6.4 \cdot 10^0$ | $-0.9 \cdot 10^0$ | $-2.9 \cdot 10^0$ | $4.0 \cdot 10^0$ | $-0.3 \cdot 10^{-9}$ | $0.5 \cdot 10^{-9}$ |
| *Std.* | $0.1 \cdot 10^0$ | $0.8 \cdot 10^0$ | $0.7 \cdot 10^0$ | $0.4 \cdot 10^0$ | $0.2 \cdot 10^0$ | $0.6 \cdot 10^0$ | $5.5 \cdot 10^{-11}$ | $4.3 \cdot 10^{-10}$ |
| P1-Atlantic (iter.7) | $-5.5 \cdot 10^0$ | $10.1 \cdot 10^0$ | $3.4 \cdot 10^0$ | $-4.5 \cdot 10^0$ | $0.7 \cdot 10^0$ | $3.5 \cdot 10^0$ | $1.7 \cdot 10^{-9}$ | $4.8 \cdot 10^{-10}$ |
| *Std.* | $0.7 \cdot 10^0$ | $1.1 \cdot 10^0$ | $2.1 \cdot 10^0$ | $1.4 \cdot 10^0$ | $1.0 \cdot 10^0$ | $3.4 \cdot 10^0$ | $8.4 \cdot 10^{-11}$ | $2.6 \cdot 10^{-10}$ |
| P1-Pacific (iter.1) | $-2.9 \cdot 10^0$ | $3.5 \cdot 10^0$ | $0.5 \cdot 10^0$ | $-0.7 \cdot 10^0$ | $-0.4 \cdot 10^0$ | $3.0 \cdot 10^0$ | $-2.5 \cdot 10^{-11}$ | $4.3 \cdot 10^{-10}$ |
| *Std.* | $1.7 \cdot 10^0$ | $4.3 \cdot 10^0$ | $0.8 \cdot 10^0$ | $0.7 \cdot 10^0$ | $1.1 \cdot 10^0$ | $1.7 \cdot 10^0$ | $2.2 \cdot 10^{-11}$ | $8.1 \cdot 10^{-10}$ |
| **Target 2** | 0 | 0 | 0 | 0 | 0 | 0 | 0 | 0 |
| P2-All (iter. 8) | $0.7 \cdot 10^0$ | $-1.8 \cdot 10^0$ | $2.9 \cdot 10^0$ | $-0.3 \cdot 10^0$ | $0.3 \cdot 10^0$ | $-1.3 \cdot 10^0$ | $0.7 \cdot 10^{-9}$ | $-1.8 \cdot 10^{-9}$ |
| *Std.* | $0.6 \cdot 10^0$ | $0.4 \cdot 10^0$ | $1.5 \cdot 10^0$ | $0.5 \cdot 10^0$ | $0.4 \cdot 10^0$ | $1.2 \cdot 10^0$ | $5.4 \cdot 10^{-11}$ | $0.4 \cdot 10^{-9}$ |
| P2-Atlantic (iter. 1) | $0.4 \cdot 10^0$ | $-3.2 \cdot 10^0$ | $1.0 \cdot 10^0$ | $1.8 \cdot 10^0$ | $-2.4 \cdot 10^0$ | $2.2 \cdot 10^0$ | $-0.2 \cdot 10^{-9}$ | $-0.1 \cdot 10^{-9}$ |
| *Std.* | $0.6 \cdot 10^0$ | $1.3 \cdot 10^0$ | $0.8 \cdot 10^0$ | $1.4 \cdot 10^0$ | $1.1 \cdot 10^0$ | $1.4 \cdot 10^0$ | $0.4 \cdot 10^{-9}$ | $0.5 \cdot 10^{-9}$ |
| P2-Pacific (iter. 6) | $-2.0 \cdot 10^0$ | $-2.8 \cdot 10^0$ | $3.1 \cdot 10^0$ | $-1.0 \cdot 10^0$ | $-0.3 \cdot 10^0$ | $-2.7 \cdot 10^0$ | $-6.4 \cdot 10^{-11}$ | $4.7 \cdot 10^{-10}$ |
| *Std.* | $0.7 \cdot 10^0$ | $4.5 \cdot 10^0$ | $3.4 \cdot 10^0$ | $0.4 \cdot 10^0$ | $0.5 \cdot 10^0$ | $1.1 \cdot 10^0$ | $8.7 \cdot 10^{-11}$ | $5.9 \cdot 10^{-10}$ |
| LGM (iter. 3) | $-2.0 \cdot 10^0$ | $5.4 \cdot 10^0$ | $6.8 \cdot 10^0$ | $-0.9 \cdot 10^0$ | $-4.8 \cdot 10^0$ | $3.5 \cdot 10^0$ | $-1.3 \cdot 10^{-9}$ | $-2.2 \cdot 10^{-10}$ |
| *Std.* | $0.4 \cdot 10^0$ | $0.7 \cdot 10^0$ | $0.9 \cdot 10^0$ | $0.3 \cdot 10^0$ | $0.2 \cdot 10^0$ | $0.5 \cdot 10^0$ | $6.1 \cdot 10^{-11}$ | $3.4 \cdot 10^{-10}$ |
| LGM (iter. 7) | $2.6 \cdot 10^0$ | $3.1 \cdot 10^0$ | $13.7 \cdot 10^0$ | $-0.7 \cdot 10^0$ | $-2.7 \cdot 10^0$ | $1.5 \cdot 10^0$ | $-7.7 \cdot 10^{-10}$ | $-4.9 \cdot 10^{-9}$ |
| *Std.* | $0.5 \cdot 10^0$ | $0.8 \cdot 10^0$ | $1.3 \cdot 10^0$ | $0.3 \cdot 10^0$ | $0.3 \cdot 10^0$ | $0.9 \cdot 10^0$ | $6.5 \cdot 10^{-11}$ | $5.9 \cdot 10^{-10}$ |



**Table 2.** *(Continued.)*

| | \multicolumn{8}{c}{Control variable} | | | | | | | |
| | 9 | 10 | 11 | 12 | 13 | 14 | 15 | 16 |
|---|---|---|---|---|---|---|---|---|
| Assumed std. | $3 \cdot 10^{-6}$ | $7 \cdot 10^{-6}$ | $3 \cdot 10^{-6}$ | $7 \cdot 10^{-6}$ | $6 \cdot 10^{-6}$ | $9 \cdot 10^{-6}$ | $5 \cdot 10^{-6}$ | $9 \cdot 10^{-6}$ |
| **Target 1** | $\mathbf{5 \cdot 10^{-6}}$ | $\mathbf{-4.0 \cdot 10^{-6}}$ | $\mathbf{7.0 \cdot 10^{-6}}$ | $\mathbf{2.0 \cdot 10^{-6}}$ | $\mathbf{-3.0 \cdot 10^{-6}}$ | $\mathbf{5.0 \cdot 10^{-6}}$ | $\mathbf{2.0 \cdot 10^{-6}}$ | $\mathbf{4.0 \cdot 10^{-6}}$ |
| P1-All (iter. 9) | $4.3 \cdot 10^{-6}$ | $-5.7 \cdot 10^{-6}$ | $4.3 \cdot 10^{-6}$ | $-3.2 \cdot 10^{-6}$ | $-4.0 \cdot 10^{-6}$ | $5.6 \cdot 10^{-6}$ | $2.2 \cdot 10^{-6}$ | $6.8 \cdot 10^{-6}$ |
| *Std.* | $5.2 \cdot 10^{-7}$ | $3.2 \cdot 10^{-6}$ | $2.2 \cdot 10^{-6}$ | $3.3 \cdot 10^{-6}$ | $2.2 \cdot 10^{-6}$ | $4.4 \cdot 10^{-6}$ | $1.9 \cdot 10^{-6}$ | $3.8 \cdot 10^{-6}$ |
| P1-Atlantic (iter.7) | $5.4 \cdot 10^{-6}$ | $-3.0 \cdot 10^{-6}$ | $2.7 \cdot 10^{-7}$ | $-4.8 \cdot 10^{-6}$ | $-4.8 \cdot 10^{-6}$ | $4.1 \cdot 10^{-7}$ | $6.7 \cdot 10^{-6}$ | $6.0 \cdot 10^{-6}$ |
| *Std.* | $7.8 \cdot 10^{-7}$ | $4.2 \cdot 10^{-6}$ | $2.8 \cdot 10^{-6}$ | $6.3 \cdot 10^{-6}$ | $2.4 \cdot 10^{-6}$ | $4.9 \cdot 10^{-6}$ | $2.7 \cdot 10^{-6}$ | $4.5 \cdot 10^{-6}$ |
| P1-Pacific (iter.1) | $-8.2 \cdot 10^{-7}$ | $2.8 \cdot 10^{-6}$ | $1.8 \cdot 10^{-6}$ | $-5.9 \cdot 10^{-7}$ | $-4.5 \cdot 10^{-6}$ | $4.2 \cdot 10^{-6}$ | $-1.6 \cdot 10^{-6}$ | $1.2 \cdot 10^{-6}$ |
| *Std.* | $2.9 \cdot 10^{-6}$ | $6.4 \cdot 10^{-6}$ | $2.5 \cdot 10^{-6}$ | $1.4 \cdot 10^{-6}$ | $3.2 \cdot 10^{-6}$ | $7.3 \cdot 10^{-6}$ | $1.9 \cdot 10^{-6}$ | $3.3 \cdot 10^{-6}$ |
| **Target 2** | $\mathbf{5 \cdot 10^{-6}}$ | $\mathbf{-4.0 \cdot 10^{-6}}$ | $\mathbf{7.0 \cdot 10^{-6}}$ | $\mathbf{2.0 \cdot 10^{-6}}$ | $\mathbf{-3.0 \cdot 10^{-6}}$ | $\mathbf{5.0 \cdot 10^{-6}}$ | $\mathbf{2.0 \cdot 10^{-6}}$ | $\mathbf{4.0 \cdot 10^{-6}}$ |
| P2-All (iter. 8) | $-1.7 \cdot 10^{-7}$ | $-1.2 \cdot 10^{-6}$ | $-7.8 \cdot 10^{-7}$ | $-1.4 \cdot 10^{-7}$ | $-5.8 \cdot 10^{-6}$ | $-1.4 \cdot 10^{-7}$ | $-2.4 \cdot 10^{-6}$ | $6.1 \cdot 10^{-7}$ |
| *Std.* | $0.6 \cdot 10^{-6}$ | $3.6 \cdot 10^{-6}$ | $2.0 \cdot 10^{-6}$ | $3.1 \cdot 10^{-6}$ | $2.2 \cdot 10^{-6}$ | $4.8 \cdot 10^{-6}$ | $2.0 \cdot 10^{-6}$ | $4.1 \cdot 10^{-6}$ |
| P2-Atlantic (iter. 1) | $-0.2 \cdot 10^{-6}$ | $-3.6 \cdot 10^{-6}$ | $-0.3 \cdot 10^{-6}$ | $-0.2 \cdot 10^{-6}$ | $-5.3 \cdot 10^{-6}$ | $4.3 \cdot 10^{-6}$ | $-1.3 \cdot 10^{-6}$ | $5.5 \cdot 10^{-6}$ |
| *Std.* | $2.0 \cdot 10^{-6}$ | $3.9 \cdot 10^{-6}$ | $2.7 \cdot 10^{-6}$ | $1.8 \cdot 10^{-6}$ | $1.8 \cdot 10^{-6}$ | $4.0 \cdot 10^{-6}$ | $3.5 \cdot 10^{-6}$ | $5.8 \cdot 10^{-6}$ |
| P2-Pacific (iter. 6) | $-5.4 \cdot 10^{-7}$ | $2.1 \cdot 10^{-7}$ | $1.4 \cdot 10^{-6}$ | $7.2 \cdot 10^{-6}$ | $-5.8 \cdot 10^{-7}$ | $7.0 \cdot 10^{-6}$ | $-1.9 \cdot 10^{-6}$ | $-2.1 \cdot 10^{-6}$ |
| *Std.* | $1.0 \cdot 10^{-6}$ | $6.6 \cdot 10^{-6}$ | $2.6 \cdot 10^{-6}$ | $3.4 \cdot 10^{-6}$ | $2.6 \cdot 10^{-6}$ | $5.7 \cdot 10^{-6}$ | $2.4 \cdot 10^{-6}$ | $4.6 \cdot 10^{-6}$ |
| LGM (iter. 3) | $-1.1 \cdot 10^{-5}$ | $-4.5 \cdot 10^{-5}$ | $-7.9 \cdot 10^{-7}$ | $-2.8 \cdot 10^{-5}$ | $-7.2 \cdot 10^{-6}$ | $-7.5 \cdot 10^{-7}$ | $-1.5 \cdot 10^{-5}$ | $-1.7 \cdot 10^{-5}$ |
| *Std.* | $6.2 \cdot 10^{-7}$ | $4.3 \cdot 10^{-6}$ | $2.0 \cdot 10^{-6}$ | $2.8 \cdot 10^{-6}$ | $2.0 \cdot 10^{-6}$ | $4.5 \cdot 10^{-6}$ | $1.7 \cdot 10^{-6}$ | $3.4 \cdot 10^{-6}$ |
| LGM (iter. 7) | $-2.5 \cdot 10^{-6}$ | $-2.5 \cdot 10^{-6}$ | $-1.0 \cdot 10^{-5}$ | $-2.7 \cdot 10^{-5}$ | $-1.3 \cdot 10^{-5}$ | $-1.4 \cdot 10^{-5}$ | $-2.0 \cdot 10^{-5}$ | $-2.0 \cdot 10^{-5}$ |
| *Std.* | $5.1 \cdot 10^{-7}$ | $2.4 \cdot 10^{-6}$ | $1.8 \cdot 10^{-6}$ | $2.7 \cdot 10^{-6}$ | $2.0 \cdot 10^{-6}$ | $3.6 \cdot 10^{-6}$ | $1.2 \cdot 10^{-6}$ | $2.7 \cdot 10^{-6}$ |





**Table 3.** Data sets utilized to constrain the LGM estimate. Number of data points refers to the available LGM-LH anomaly data points.

| Data type (LGM-LH anomalies) | Area | # Data points | Data compilation |
|---|---|---|---|
| SST | Global Ocean | 667 | MARGO Project Members (2009) |
| Planktic $\delta^{18}O_c$ | Global Ocean | 136 | Waelbroeck et al. (2014) |
| | Global Ocean | 114 | Caley et al. (2014) |
| | Global Ocean | 123 | This study (LGM) and Waelbroeck et al. (2005) (LH) |
| Benthic $\delta^{18}O_c$ | Atlantic Ocean | 163 | Marchal and Curry (2008) |
| | Global Ocean | 114 | Caley et al. (2014) |
| | Atlantic Ocean | 5 | Völpel et al. (2018) |



**Table 4.** Cost function values in pseudo-proxy and LGM experiments. For each experiment the iteration with the lowest model-data misfit ($J_{\mathrm{misfit}}$) is given, except for additional iterations for P1-Pacific (iteration 6) and the LGM experiment (iteration 7). The terms $J'_{\mathrm{misfit}}$ and $J'_{\mathrm{ctrl}}$ denote the normalized model-data misfit and normalized deviation from the first guess control variables, that is, $J_{\mathrm{misfit}}$ divided by the number of model-data comparisons and $J_{\mathrm{ctrl}}$ divided by the number of control variables. Respective $J'_{\mathrm{misfit}}$ values for the SST, planktic, and benthic $\delta^{18}O_c$ data are given. In the pseudo-proxy experiments the columns for planktic and benthic $\delta^{18}O_c$ refer to surface and deep ocean $\delta^{18}O_{sw}$ sampled at the location of the planktic and benthic $\delta^{18}O_c$, respectively. According to the theory of a $\chi^2$-test, a value of one indicates model-data agreement within the uncertainties of the observations ($J'_{\mathrm{misfit}}$) or change of the control variables within the estimated uncertainty ($J'_{\mathrm{ctrl}}$). Numbers in brackets provide the cost function values for the global data.

| Experiment | Proxy data coverage | $J'_{\mathrm{misfit}}$ | | | $J'_{\mathrm{ctrl}}$ | AMOC |
| | | SST | Planktic $\delta^{18}O_c$ | Benthic $\delta^{18}O_c$ | | |
|---|---|---|---|---|---|---|
| **Target 1** | | | | | | **12.3 Sv** |
| First guess | All | 1.5 | 15.4 | 2.9 | 0 | 17.2 Sv |
| P1-All (iter. 9) | All | 1.0 | 1.0 | 1.0 | 0.8 | 12.2 Sv |
| P1-Atlantic (iter. 7) | Only Atlantic | 1.0 (1.3) | 1.1 (1.2) | 1.0 (1.0) | 1.1 | 13.3 Sv |
| P1-Pacific (iter. 1) | Only Indian & Pacific Ocean | 1.0 (1.2) | 1.0 (1.4) | 1.0 (1.0) | 0.2 | 15.3 Sv |
| P1-Pacific (iter. 6) | Only Indian & Pacific Ocean | 1.0 (1.3) | 1.0 (11.1) | 1.2 (2.1) | 0.3 | 0.5 Sv |
| **Target 2** | | | | | | **13.6 Sv** |
| First guess | All | 1.1 | 6.3 | 1.7 | 0 | 17.2 Sv |
| P2-All (iter. 8) | All | 1.0 | 1.0 | 1.0 | 0.4 | 14.1 Sv |
| P2-Atlantic (iter. 1) | Only Atlantic | 1.1 (1.1) | 1.1 (1.2) | 1.0 (1.0) | 0.2 | 16.6 Sv |
| P2-Pacific (iter. 6) | Only Indian & Pacific Ocean | 1.0 (1.6) | 1.0 (6.4) | 1.0 (1.4) | 0.2 | 0.5 Sv |
| **LGM proxy data** | | | | | | |
| First guess | | 1.2 | 41.6 | 3.9 | 0 | 17.2 Sv |
| LGM (iteration 3) | | 1.3 | 9.3 | 2.7 | 5.7 | 13.8 Sv |
| LGM (iteration 7) | | 1.3 | 9.6 | 2.9 | 5.6 | 0.4 Sv |