# Peer review of "A reduced-order Kalman smoother for (paleo-)ocean state estimation: assessment and application to the LGM"

_Geoscientific Model Development, 2019_

## Referee Comment (RC1) · Anonymous Referee #1 · 14 Jun 2019

**Review of "A reduced Kalman Smoother for (paleo-)ocean State Estimation: Assessment and Application to the LGM" by Breitkreuz et al. 2019**

**General Comments**

In paleoclimate simulations, General Circulation Models (GCM) are normally used to conduct equilibrium simulations. In this way the model state is highly dependent from model parameters and boundary conditions. The authors propose a method to constrain the atmospheric forcings of an Ocean General Circulation Model (OGCM) with proxy reconstructions, in order to derive past ocean states, specifically for the Last Glacial Maximum (LGM). The authors propose a state reduction approach together with a finite difference sensitivity Kalman smoother to estimate forcing fields constrained by proxy data.

The study is believed to be interesting for the paleoclimate modeling community. Nevertheless, in its current form the manuscript suffers from a series of major issues that need to be addressed before it may be considered for publication. As a general very important comment, the paper text and structure are very poor and really difficult to read. Many parts are not carefully described. The order of paragraphs, figures and tables often does not follow a consistent flow, with things being mentioned before their description. Many technical features should be better and more clearly described. The grammar should also be revised. Beside this general comments, a series of more specific issues also need to be carefully considered.

**Specific comments**

- The contents of the introduction are not always well connected. Consequently, the text is not easily readable. Additionally, a more detailed review of other paleo-data assimilation applications is missing. I think some more words on the motivation for applying data assimilation to paleoclimate studies could also be added.

- You dedicate a long paragraph of your introduction to the adjoint method. From your text it seems like the adjoint method is the only data assimilation method used for paleoclimate applications. This also happens in other sections. Other paleo-data assimilation applications with using different methods should be considered in my opinion in the introduction, for then focusing on the adjoint method.

- The Order of the subsections in the methods section should be revised. In particular, you should consider moving the "LGM Proxy Data and Uncertainties" subsection after the model one, at the beginning. In fact, when you describe your experiments you need to know what are the points you consider

in your experiments, and this is only possible once you introduced the proxies and their distribution. This will make the reading of the methods more easily understandable.

- You applied your method to a set of 16 control variables. You justify this selection saying that they "have a large influence on the simulated ocean circulation and on the simulated oxygen isotopic distribution". Can you objectively quantify their influence, in the model, in particular in comparison to other variables?

- When you apply the Finite Difference Sensitivity-Iterative Kalman Smoother, in order to calculate the Jacobian of the model $G_l$ around the control variables $\lambda_l$, you use a linear regression for each control variable with the 3+1 sensitivity experiments and corresponding model outputs. Can you show how good is a linear regression for each of the control variables? What happens when the model response to variable perturbations is highly non-linear? In this case, does it make sense to use your method for those variables?

- It would be interesting to see what happens in experiment 3 (the one using proxy data), when you randomly change the total number of proxies used.

- If you want to consider the AMOC strength as a measure for success in reconstructing the ocean circulation, it seems that your approach is not really successful. At least, your results do not support such strong conclusions. In particular, it seems that your results depend, among several other factors, on the iteration considered in each case. I would suggest to revise any strong conclusion on the "success" of the method throughout the text, being more careful.

- It is not clear what are the conclusions arising from the first two set of experiments. In particular, you design those experiments for evaluating your method when the model does not include any error beside the one in the control variables and when it does. What can you say in this respect? You discuss the results of these two sets of experiments in section 4.2. But here you mainly address the data coverage constraint. Why could you not also use the proxy data for answering this question? It would also be really interesting to see what happens when you use the proxy data only for specific areas. In general, there is no discussion on the reasons for differences among the sets of experiments 1 and 2. More attention should be devoted to this point and to the evaluation of the values obtained for the control variables in the two cases.

- I highly recommend to present the results of table 3 in a more direct readable way. Maybe a graphical representation of the results would help in this case. I would also suggest to present the results obtained for all the iterations, maybe in the supplements. I think this could help in the interpretation of changes in the cost function. One thing that is not mentioned in the text is the plausible range of values that each control variable might assume. I think this is really

important in order to understand the changes presented in table 3.

- For evaluating the results of the simulations using proxy data, do you have any estimate of what should be the target LGM AMOC strength, as derived from other studies?

- In the discussion part you highlight several controversial aspects of your method, that would not make it work really efficiently. These are the first guess dependency of the results, the non normality of the observational errors and the non-linear effects originating from the combinations among different variables. You also mention as a possible cause of problems the scarce number of proxy locations for the LGM. All these issues, in my opinion, require more attention in the paper in order to may properly evaluate the reliability of your method. For example, you could investigate the effects of the number of proxies on the results by conducting additional pseudo-proxy experiments increasing their number.

- Conclusions should be expanded and in some parts reformulated accordingly to the suggested modifications.

**Minor Comments**
- Has your State reduction approach ever been used for other climate applications. Emphasize this point in the introduction.

- Page 2, line 4: You give information on the importance of LGM. But how is it connected with your first sentence?

- Page 2, line 13: Explain what is the adjoint method briefly.

- Page 2, line 15: correct "estimated that correspond" into "estimated, corresponding"

- Page 2, Line 26/27: Reference missing

- Page 3, line 1: Acronyms with first capital letter

- Page 3, line 29: with a varying resolution of

- Page 4, line 22: specify that 73728 is the number of variables per field per year.

- Page 5, lines 17-18: Which kind of smoother did you use?

- Page 7, line 10-12: This part is not really clear. You refer to the estimation of the Kalman filter?

- Page 8, line 12: 10 iterations

- Page 8, lines 16-31: There are several inconsistencies in verb tenses: try to correct them.

- Page 8, line 27: "it is hardly ever fullfilled": references needed

- Page 8, line 28: Can you add a figure for showing the distribution of the data?

- Page 11, line 18: The FDS-IKS is very successful... : check again consistency of verb tenses

- Page 11, line 26: what do you mean when you say that the variables were adjusted in the wrong direction? what would be the correct one? are they too large or too small? please be more specific

- Page 12, line 22: indicating a perfect model-data agreement: You can say that is perfect because you apply a chi-squared test? eventually specify it.

- Figure 3,4: Use different colours for the different realizations, since the ones you use now to draw lines do not allow to easily distinguish among them

- Table 2: substitute the numbers of the control variables with their names, in the top of the table

---

## Referee Comment (RC2) · James Annan (Referee) · 4 Jul 2019

For a number of reasons I've found it difficult to obtain two reviews for this manuscript. To avoid further delays I am taking the slightly unusual step of acting as reviewer despite also being Editor.

The manuscript presents an interesting method for parameter estimation in an ocean model, with application to paleoclimate. The paper is in scope for GMD and with some revision it could potentially be published.

My main concern is with the details of the finite difference approximation, which seems

a bit arbitrary (at least based on the presentation). Why were three perturbations used, and selected randomly? The size of sigma/100 seems very small. There is clearly a compromise here due to model drift/disequilibrium/internal variability which will vary with application. Larger perturbations would provide a more robust estimate and if the response is nonlinear then wouldn't it be better to use a more realistic estimate of the average response over a larger increment anyway, since you are in fact applying substantial increments in your optimisation? If you can really get away with a small finite difference approximation then even a single perturbation would have sufficed, but the information from smaller and large perturbations could contain useful information regarding the nonlinearity (and/or the accuracy of the scheme in estimating the true response). For example, with 3 samples, you could have chosen perturbations of +-sigma/2 and +sigma/100 which would have given both the local tangent and also some indication as to the range of its validity. For a journal such as GMD I'd like to see more analysis of how the method works, ie how collinear the sample points are, and also how well the predicted optimum performs in comparison to expectation at the update time.

The comparison to an adjoint is a little harsh in places. There are various ways of ensuring smooth forcing fields with that method, including the state space reduction you have applied here, or also some regularisation as an additional constraint on the forcing fields. An additional limitation of the method you have presented here is that in the case of strong covariances between your control variables, any optimisation may be very slow and inefficient as you are only evaluating the orthogonal directions (interactions would require $O(p^2)$ simulations). Therefore, the choice of control variables is probably very critical and the method will have limited applicability.

The state space reduction is a good idea, I'm not sure to what extent it may be novel in this area. It would be useful to have a little more in the way of references and/or discussion here (eg explaining the choice of Legendre polynomials).

In Figure 1 (left) I'd like to also see the optimised results which could be presented as

deviations from target in the case they are invisibly small.

I think the results in Table 2 might be more readable if presented graphically, perhaps some sort of scatter plot after nondimensionalisation. Eg deviations from the true value. You could use colours, symbols, line types etc to distinguish them.

———————————————